# Blockage of autophagy causes severe skeletal muscle disruption in a mouse model for myofibrillar myopathy 6

Kerstin Filippi[1], Kathrin Graf-Riesen[1], Maithreyan Kuppusamy [2,3], Andreas Unger[4], Kenichi Kimura [5], Martin Matijass[2], Henrique Baeta[2,6], Magdalena Podlacha [7], Daniel Haertter [8], Alexei P. Kudin[9,10], Martin Wiemann[1], Grzegorz Węgrzyn [7], Cornelia Kornblum [11], Jens Reimann[11], Wolfgang A. Linke [4], Pitter F. Huesgen [2,6,12], Wolfram S. Kunz [9,10], Bernd K. Fleischmann [1] & Michael Hesse [1] ✉

Myofibrillar myopathy 6 is a rare, autosomal-dominant neuromuscular disorder caused by an amino acid exchange Pro209Leu in the co-chaperone *BAG3*, which disrupts muscle protein turnover and causes severe muscle weakness and shortened lifespan. We generated transgenic mice overexpressing the human mutant BAG3[P209L]-GFP, which rapidly develop skeletal muscle weakness unlike controls expressing BAG3[WT]-GFP. Here we show that mutant mice exhibit sarcomere breakdown, inflammation, protein aggregates, centralized nuclei and mitochondrial defects in their skeletal muscles, thereby reducing contraction force by ~90%. Omics profiling uncovered impaired protein synthesis, blocked autophagy, impaired mitophagy and loss of sarcomere proteins. Pathway modulation in vitro and in vivo showed autophagy dysfunction as the primary driver for the pathology, while BAG3 knockdown gene therapy markedly restored muscle function in vivo. In summary, this model recapitulates core disease features, revealing how BAG3 aggregates and loss of BAG3 function impair autophagy to drive muscle degeneration.

Myofibrillar myopathies (MFM) are defined by disintegration of myofibrils and aggregation of proteins into intracellular inclusions and manifest as progressive skeletal muscle weakness. While the majority of MFMs are caused by mutations in structural genes, such as *TTN* or *DES*, some are caused by mutations in genes encoding protein quality control factors, such as *BAG3* (Bcl-2 associated athanogene 3).

BAG3 is a co-chaperone that forms the chaperone-associated selective autophagy (CASA) complex together with HSPB8, HSPA8, and CHIP. CASA is able to bind to misfolded sarcomeric proteins, such as Filamin C, and trigger their degradation by autophagy[1]. Because of this role of BAG3 as a central node in protein quality control, it may affect many sarcomeric proteins and the maintenance of myocyte

[1]Institute of Physiology I, Medical Faculty, University of Bonn, Bonn, Germany. [2]Central Institute for Engineering, Electronics and Analytics, ZEA-3, Forschungszentrum Jülich, Jülich, Germany. [3]Department Metabolism, Senescence and Autophagy, Research Center One Health Ruhr, University Alliance Ruhr & University Hospital Essen, University Duisburg–Essen, 45147 Essen, Germany. [4]Institute of Physiology II, University of Muenster, Muenster, Germany. [5]Life Science Center for Survival Dynamics, Tsukuba Advanced Research Alliance (TARA), University of Tsukuba, Ibaraki 305-8577, Japan. [6]Institute of Biology II, University of Freiburg, 79104 Freiburg, Germany. [7]Department of Molecular Biology, Faculty of Biology, University of Gdańsk, Gdańsk, Poland. [8]Institute of Pharmacology and Toxicology, University Medical Center Göttingen, Göttingen, Germany. [9]Institute of Experimental Epileptology and Neurocognition, University Bonn Medical Center, 53105 Bonn, Germany. [10]Department of Epileptology, University Bonn Medical Center, 53105 Bonn, Germany. [11]Department of Neurology, University Bonn Medical Center, 53105 Bonn, Germany. [12]CIBSS-Centre for Integrative Biological Signaling Studies, University of Freiburg, 79104 Freiburg, Germany. ✉e-mail: mhesse1@uni-bonn.de

function. A point mutation in the BAG3 gene (c.626 C > T, p.P209L) causes BAG3[P209L]-associated myofibrillar myopathy (MFM6), which is a severe and potentially life-threatening condition. Affected patients experience proximal and generalized muscle weakness that progresses rapidly, polyneuropathy, and sometimes restrictive cardiomyopathy with early onset during childhood[2]. The primary cause of death in these patients is often respiratory failure due to skeletal muscle weakness, and they have a life expectancy of around 20 years. Histological hallmarks of the disease are the formation of cytoplasmic protein inclusions, disintegration of sarcomeres, and Z-line streaming[2].

Several attempts were undertaken to generate a mouse model that mimics the cardiac and skeletal muscle phenotypes of patients. Beginning with the overexpression of human BAG3[P209L] in mouse cardiomyocytes, some aspects of the cardiac pathology were recapitulated[3]. Introduction of the homolog mutation (P215L) into mouse *Bag3* did not lead to an obvious phenotype even when mice were bred to homozygosity[4]. A compound line consisting of a Bag3[P215L] knock-in and a Bag3 knock-out caused a mild skeletal muscle phenotype but without any impairment in muscle strength[5]. To better mimic the cardiac and skeletal muscle pathologies observed in patients, we developed a novel humanized mouse model for MFM6 in which a fusion protein of human BAG3[P209L] and eGFP is ubiquitously expressed[6].

In the hearts of these mice, the solubility of BAG3 is altered due to the point mutation that renders the protein more prone to aggregation[6,7]. Accordingly, BAG3 is no longer able to fulfill its function as a co-chaperone in the CASA complex, and its loss of function leads to the accumulation of damaged muscle proteins and consequently to the disintegration of sarcomeres. Besides the severe cardiac phenotype, we noticed that these mice display clear signs of muscle weakness and are therefore an ideal model for studying the pathomechanism underlying MFM6 in skeletal muscle. Since skeletal muscle is a tissue capable of regeneration by tissue-derived stem cells, it is particularly interesting to explore the differences between skeletal and cardiac muscle, as the latter lacks stem cells and exhibits poor regeneration.

The analysis of our mouse model revealed a striking decrease in muscle force and sarcomere disintegration, along with protein aggregate formation and accumulation of autophagy and mitophagy markers. Since skeletal muscle is a regenerative tissue, we observed an increase in protein synthesis in the transgenic mice, but this was not sufficient to compensate for the loss of sarcomere proteins. Additionally, mitochondrial function was severely impaired and mitophagy was blocked, potentially contributing to the skeletal muscle phenotype. Notably, induction of autophagy in vivo and in vitro, but not induction of mitophagy, was able to attenuate the phenotype. Taken together, our mouse model mimics the skeletal phenotype of MFM6 and provides new insights into disease mechanisms mainly caused by a blockage of autophagy and can be rescued by an shRNA-mediated knockdown of BAG3[P209L].

## Results

To investigate the pathomechanism of BAG3[P209L]-associated MFM, we generated a humanized mouse model with inducible overexpression of human BAG3[P209L] fused to eGFP. In addition, to rule out any effects caused by BAG3-eGFP overexpression, we generated a control mouse line overexpressing human BAG3 fused to eGFP[6]. Both inducible mouse lines were crossed with PGK-Cre transgenic mice to achieve ubiquitous expression of BAG3[P209L]-eGFP and BAG3[WT]-eGFP, respectively, and then further bred to homozygosity (homozygous PGK-Cre/CAG-flox-hBAG3[P209L]-eGFP mice are referred to as CAG-BAG3[P209L], and homozygous PGK-Cre/CAG-flox-hBAG3[WT]-eGFP mice as CAG-BAG3[WT]).

CAG-BAG3[P209L]-mice were significantly smaller than their control siblings and exhibited typical signs of muscle wasting, such as rigid spine and malalignment of the hind legs resulting in a waddling gait

(Fig. 1a–c). In contrast, CAG-BAG3[WT]-mice showed no obvious phenotype and were indistinguishable from non-transgenic littermate control (CTRL) mice (Supplementary Fig. 1a–k). Life expectancy of CAG-BAG3[P209L]-mice was about 5–8 weeks, as reported before[6], due to the restrictive cardiomyopathy leading to heart failure. The mice did not lose weight but grew more slowly than their control littermates (Supplementary Fig. 2a) and CAG-BAG3[WT]-mice (Supplementary Fig. 1a), indicating growth restriction. In fact, e.g., both the M. soleus and also the EDL were clearly smaller than in controls (Fig. 1d), and statistical analysis of muscle length and weight confirmed significantly shorter and lighter EDL and soleus muscles in 2- and 4-week-old CAG-BAG3[P209L]-mice (Supplementary Fig. 2b, c). EGFP expression was detected in the muscles of these mice (e.g. soleus muscle and diaphragm), which was not seen in CTRLs (Fig. 1e, f) but was observed in CAG-BAG3[WT]-mice (Supplementary Fig. 1b). Cross-sections through the EDL, soleus, quadriceps muscles, or diaphragm revealed expression of hBAG3[P209L]-eGFP or hBAG3[WT]-eGFP (Fig. 1f, g; Supplementary Fig. 1c). Despite clear differences in muscle size and weight, we found that tibia length at 5 weeks of age was not significantly reduced in CAG-BAG3[P209L]-mice, indicating muscle atrophy (Supplementary Fig. 2d). To get an impression of the extent of overexpression, we measured the amount of mBag3 and hBAG3 by Western Blot analysis. We found that the total amount of BAG3 protein (the sum of endogenous Bag3 and transgenic BAG3-eGFP) in quadriceps muscle from 4-week-old CAG-BAG3[P209L] and CAG-BAG3[WT] mice was $11.36 \pm 0.45$ times higher in CAG-BAG3[P209L] mice (Supplementary Figs. 2e) and $4.46 \pm 0.04$ times higher in CAG-BAG3[WT] mice (Supplementary Fig. 3a) than in CTRL mice (Supplementary Fig. 2f). To determine whether this increase was due to elevated mRNA levels or accumulation of protein as a manifestation of the phenotype, we next measured the amount of transgenic hBAG3[P209L]-eGFP and endogenous mouse *Bag3* mRNA by qPCR with probes specific for mouse and human BAG3. The relative mRNA-levels of mouse *Bag3* normalized to GAPDH were not statistically significantly different (Supplementary Fig. 2g). The same was true for the expression of the human *BAG3* transgene mRNA in CAG-BAG3[P209L] and CAG-BAG3[WT]-mice (Supplementary Fig. 2g). Since the total mRNA level of *BAG3* (mouse *Bag3* + human *BAG3*) induction was only $3.01 \pm 1.41$ and $3.37 \pm 1.77$ compared to CTRL mice (Supplementary Fig. 2h), this indicated an accumulation of mBag3/BAG3 protein, which was very prominent in CAG-BAG3[P209L] mice but not in CAG-BAG3[WT] mice. However, compared to skeletal muscle, total BAG3 protein was $3.01 \pm 0.69$ times more abundant in heart muscle from CAG-BAG3[P209L]-mice (Supplementary Fig. 3a). Also, the ratio of human BAG3-eGFP protein to endogenous mouse Bag3 protein was $3.56 \pm 0.20$ (Fig. 5n) which was higher than in cardiac muscle, because the amount of mouse Bag3 protein was lower in skeletal muscle (Supplementary Fig. 3b, c). In summary, the increase in total *BAG3* mRNA (mouse *Bag3* + human *BAG3*) was significantly lower than the increase in total BAG3 protein in skeletal muscle in CAG-BAG3[P209L]-mice, which is likely a sign of the pathological phenotype, probably reflecting protein aggregation.

## Centralized nuclei and fiber type switch in skeletal muscle from CAG-BAG3[P209L] mice

For histological analysis of skeletal muscle, we focused on quadriceps femoris muscle and the extensor digitorum longus (EDL) muscle as muscles with a high percentage of fast-twitch fibers, and on *M. soleus* as a muscle with a high percentage of slow-twitch fibers. Since respiratory muscle failure is a major cause of death in patients, we also examined the diaphragm. Similar to other muscular dystrophies, we found a significant increase in the number of centralized myocyte nuclei in all four muscles compared to controls, indicating a regenerative response (Fig. 1i–p). This was not seen in CAG-BAG3[WT]-mice (Supplementary Fig. 1d, e). In addition, in skeletal muscle of CAG-BAG3[P209L] mice there was an accumulation of degenerating muscle cells and general atrophy of muscle fibers, which was evident by their smaller size in cross

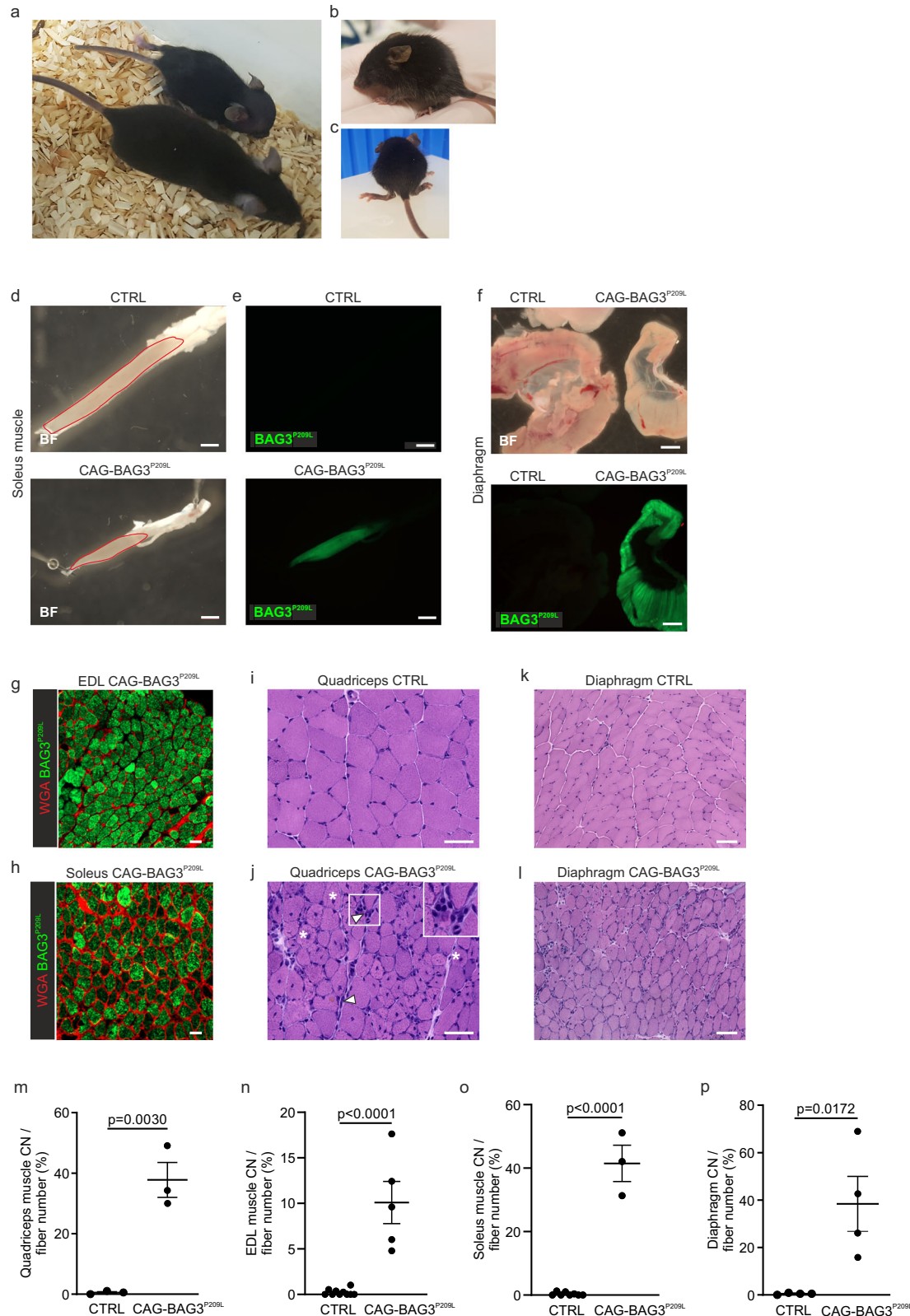

sections (Fig. 1j, l; Supplementary Fig. 3d). Muscle fibers were infiltrated by interstitial cells, which were likely fibroblasts or immune cells (Fig. 1j and Supplementary Fig. 3d, arrowheads). As fibrosis is a main feature of the phenotype in hearts of CAG-BAG3[P209L]-mice[6], we stained sections of EDL, soleus, and diaphragm muscles for collagen by Sirius Red (Fig. 2a) and quantified the fibrotic area (Fig. 2b–d). There was a significant increase in fibrotic tissue in the EDL, soleus and diaphragm

muscles from CAG-BAG3[P209L]-mice, but not in CAG-BAG3[WT]-mice (Supplementary Fig. 1 f,g), and it was not as prominent as in hearts from these mice[6]. The number of satellite cells in the EDL and soleus muscles was unchanged in CAG-BAG3[P209L]-mice compared to controls (Supplementary Fig. 3e–g), indicating no impairment of regeneration due to stem cell pool depletion. Despite using the ubiquitous CAG promoter, BAG3[P209L]-eGFP was not expressed in satellite cells.

**Fig. 1 | Muscular dystrophy in CAG-BAG3[P209L]-mice. a** CAG-BAG3[P209L]-mice were smaller than their non-transgenic littermates. **b, c** Typical signs of muscle wasting such as rigid spine (**b**) and malalignment of the hind legs, resulting in a waddling gait (**c**). **d** Significantly smaller soleus muscle of CAG-BAG3[P209L]-mice compared with control mice. **e, f** Homogenous BAG3[P209L]-eGFP expression in soleus muscle and diaphragm from CAG-BAG3[P209L]-mice. **g, h** Strong eGFP fluorescence in cross-sections from EDL (**g**) and soleus muscle (**h**) from CAG-BAG3[P209L]-mice. **i–l** Hematoxylin-eosin staining demonstrated centralization of myocyte cell nuclei in quadriceps muscle (**j**) and diaphragm (**l**) from 4 weeks old CAG-BAG3[P209L]-mice (asterisks) and accumulation of infiltrating cells (arrowheads, insert shows an enlargement of infiltrating interstitial cells) but not in controls (**i, k**). Quantification revealed a significant increase of centralized nuclei in quadriceps muscle (**m**) ($n = 3$ biologically independent mice per group), EDL muscle (**n**) ($n = 5$ biologically independent CAG-BAG3[P209L] mice; $n = 9$ biologically independent CTRL mice), soleus muscle (**o**) ($n = 3$ biologically independent CAG-BAG3[P209L] mice; $n = 8$ biologically independent CTRL mice), and diaphragm (**p**) ($n = 4$ biologically independent mice per group) of CAG-BAG3[P209L]-mice, shown is mean ± SEM, statistical test = two-sided unpaired Student's $t$-test. Scale bars = 1000 μm (**d–f**), 25 μm (**g, h**), and 50 μm (**i–l**). **g–p** All experiments were repeated three times using biologically independent replicates, yielding similar results. Source data are provided as a Source Data file.

A change in muscle fiber types is often observed in muscular dystrophies that have concomitant neuropathies, as observed in MFM6 patients. Interestingly, we noticed a switch in skeletal muscle fiber types from fast-twitch to slow-twitch types in 4-week-old CAG-BAG3[P209L] mice (Supplementary. Fig. 4a–d). In both soleus and EDL muscles, the percentage of MHC-I type fibers increased, while the percentage of MHC-IIb fibers decreased (Supplementary Fig. 4e, f). Additionally, MHC-IIb fibers in the EDL muscle of CAG-BAG3[P209L]-mice had a smaller diameter (Supplementary Fig. 4g), indicating atrophy, while in soleus muscle this was unchanged (Supplementary Fig. 4h). There was no correlation between the level of hBAG3-eGFP expression and fiber type in either soleus or EDL muscle at 4 weeks (Supplementary Fig. 4i, j) of age. The change in the composition of fiber types indicated a neuronal phenotype in our mouse line, which is also observed in MFM6 patients.

## Functional impairment and aggregate formation in skeletal muscle from CAG-BAG3[P209L] mice

Since skeletal muscles were smaller, we wondered whether muscle function was decreased and therefore measured force development in CAG-BAG3[P209L]-mice and controls. We performed frequency-dependent force measurements (FFR) with isolated EDL and soleus muscles from 4-week-old CAG-BAG3[P209L], CAG-BAG3[WT], and control mice. Muscles were stimulated with electric pulses at increasing frequencies from 10 to 125 Hz (Fig. 2d–g, Supplementary Fig. 1h, i). In individual muscles, maximal force was reduced by ~3-fold (Fig. 2d, e). To exclude that this was due to different muscle size, the maximal force was normalized to muscle length and weight. In both muscles, the maximal normalized force was significantly (1.5 to 2-fold) decreased in CAG-BAG3[P209L]-mice. This was especially evident at high stimulation frequencies (Fig. 2f, g), which induce fused tetanic contractions and did not occur in CAG-BAG3[WT]-mice (Supplementary Fig. 1h, i). Interestingly, the decrease in maximal force negatively correlated with the intensity of CAG-BAG3[P209L]-eGFP expression, which displayed some degree of variability among the transgenic mice (Supplementary Fig. 4k, l). As maximal force was normalized, it became apparent that the reduced force was due to either structural changes, such as the disintegration of sarcomeres, or to an impairment in energy metabolism, or to both.

Because a hallmark of MFM6 is the formation of cytoplasmic protein inclusions in skeletal muscle[2], we performed an immuno-fluorescence analysis of sections from the EDL and soleus muscles of 4-5-week-old CAG-BAG3[P209L]-mice. BAG3[P209L]-eGFP protein formed cross-striated structures in the muscles from control mice (Fig. 3a–d, arrows) and CAG-BAG3[WT]-mice (Supplementary Fig. 1j, k), while in muscles from BAG3[P209L]-mice, irregular structures and protein aggregates were found (Fig. 3a–d, arrowheads). Staining for ACTN2 revealed the typical cross-striation of sarcomeric Z-disks, which was disturbed and severely impaired in myofibers from CAG-BAG3[P209L]-mice (Fig. 3a, b) but not CAG-BAG3[WT]-mice (Supplementary Fig. 1j). The desmin intermediate filaments displayed abnormalities and a loss of their regular structure when compared with controls and CAG-BAG3[WT]-mice (Fig. 3c, d; Supplementary. Fig. 1k), hinting at decreased integrity of the myocyte IF

cytoskeleton, as has been reported in mice carrying a dominant negative desmin mutation[8]. A potential block of autophagy was assessed by quantification of SQSTM1 and LC3B stainings (Fig. 3e, f). SQSTM1 and LC3B accumulated in small puncta in the quadriceps muscle of CAG-BAG3[P209L] mice (Fig. 3e), and quantitation revealed a statistically significant increase in both values (Fig. 3f, g). Since protein aggregates accumulated in skeletal myocytes of these mice, the increase in SQSTM1 is due to a blockage of autophagy and not to an increase.

To assess the degree of intercellular damage in skeletal muscle, parts of the quadriceps femoris muscle were dissected from CAG-BAG3[P209L] and control mice, and ultrathin sections were prepared for ultrastructure analysis by electron microscopy. In the electron micrographs, the skeletal muscles of controls showed a regular arrangement of sarcomeres with typical transverse striation (Fig. 3h), whereas in skeletal muscles from CAG-BAG3[P209L]-mice, disintegration of sarcomeres with intersarcomeric electron-dense regions (Fig. 3i, arrows) was evident. This is a typical finding in myofibrillar myopathies and one of the hallmarks of the human pathology[9]. Various types of aggregates (granulo-filamentous and vesiculated-subsarcolemmal) could be observed. Numerous lysosomal and autophagic vacuoles were also visible, indicating increased autophagy or a blockage of autophagy manifested by an accumulation of autophagosomes (Fig. 3j). Notably, mitochondria were disarranged in skeletal muscle from CAG-BAG3[P209L]-mice and displayed a variety of shapes and sizes compared to controls (Fig. 3i, arrowheads). Signs of mitophagy, such as remnants of mitochondrial cristae in autophagosomes, were also visible (Fig. 3j, asterisks). Overall, the structural changes resembled those described in patients with MFM6.

Loss of structural integrity could lead to cell decay followed by an immune response. Therefore, we screened for both inflammatory markers and programmed cell death markers. There was a significant increase in CD45-positive cells in the quadriceps and both EDL and soleus muscles from CAG-BAG3[P209L]-mice (Fig. 3k, l, Supplementary Fig. 5a), as a sign of immune cell infiltration and inflammation, which was confirmed by increased staining for acidic phosphatase (Supplementary Fig. 5b). In addition, staining for cleaved caspase 3 revealed a statistically significantly higher number of apoptotic fibers in both EDL and soleus muscles (Supplementary Fig. 5c, d). These apoptotic muscle fibers were likely replaced by fibroblasts, which produced more extracellular matrix proteins, as Gomori Trichrome staining revealed an increase in collagen deposition in CAG-BAG3[P209L]-mice (Supplementary Fig. 5e).

Staining for the cell cycle marker Ki-67 was performed to obtain first information about possible satellite cell proliferation in the skeletal muscles of CAG-BAG3[P209L]-mice (Supplementary Fig. 5f). A significant difference was observed between CTRL and CAG-BAG3[P209L]-mice in the soleus muscle but not in the EDL muscle (Supplementary Fig. 5f, g), indicating increased cell proliferation during regenerative processes in some skeletal muscles of CAG-BAG3[P209L]-mice. However, the underlying molecular mechanisms causing this chain of events were unclear, so we performed Omics analysis to gain deeper insights.

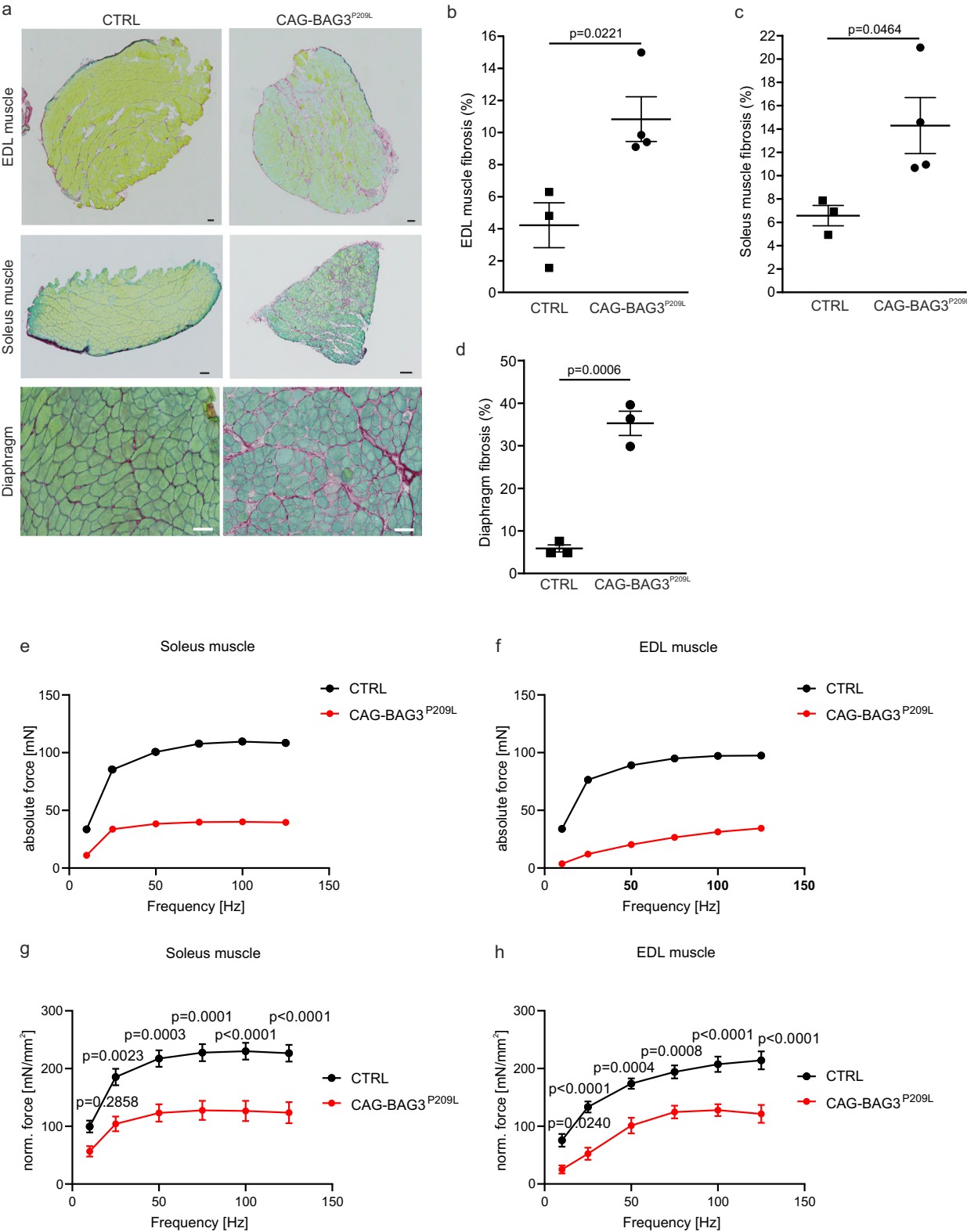

**Fig. 2 | Mild fibrosis and loss of force in muscles from 4 weeks old CAG-BAG[P209L]-mice. a** Sirius Red-staining with Fast Green counterstain of EDL, soleus and diaphragm muscles from 4 weeks old CAG-BAG3[P209L] and control mice. Data are representative of three experiments with biologically independent replicates. Scale bars = 100 μm. **b–d** Quantification of fibrosis in soleus (*n* = 4 biologically independent CAG-BAG3[P209L] mice; *n* = 3 biologically independent CTRL mice), EDL (*n* = 4 biologically independent CAG-BAG3[P209L] mice; *n* = 3 biologically independent CTRL mice), and diaphragm (*n* = 3 biologically independent mice per group) muscles from (**a**); shown is mean ± SEM, statistical test = two-sided unpaired Student's *t*-test.

**e–h** EDL and soleus muscle of four-week-old CAG-BAG3[P209L] mice develop less force than control mice during frequency-dependent force measurement (FFR) before (**e**, **f**) and after normalization (**g**, **h**). Sol.: *n* = 6 biologically independent CAG-BAG3[P209L] mice; *n* = 4 biologically independent CTRL mice; EDL: *n* = 5 biologically independent CAG-BAG3[P209L] mice; *n* = 6 biologically independent CTRL mice. Shown is mean ± SEM, statistical test = two-way repeated-measures ANOVA, followed by Šídák's multiple-comparisons test. Source data are provided as a Source Data file.

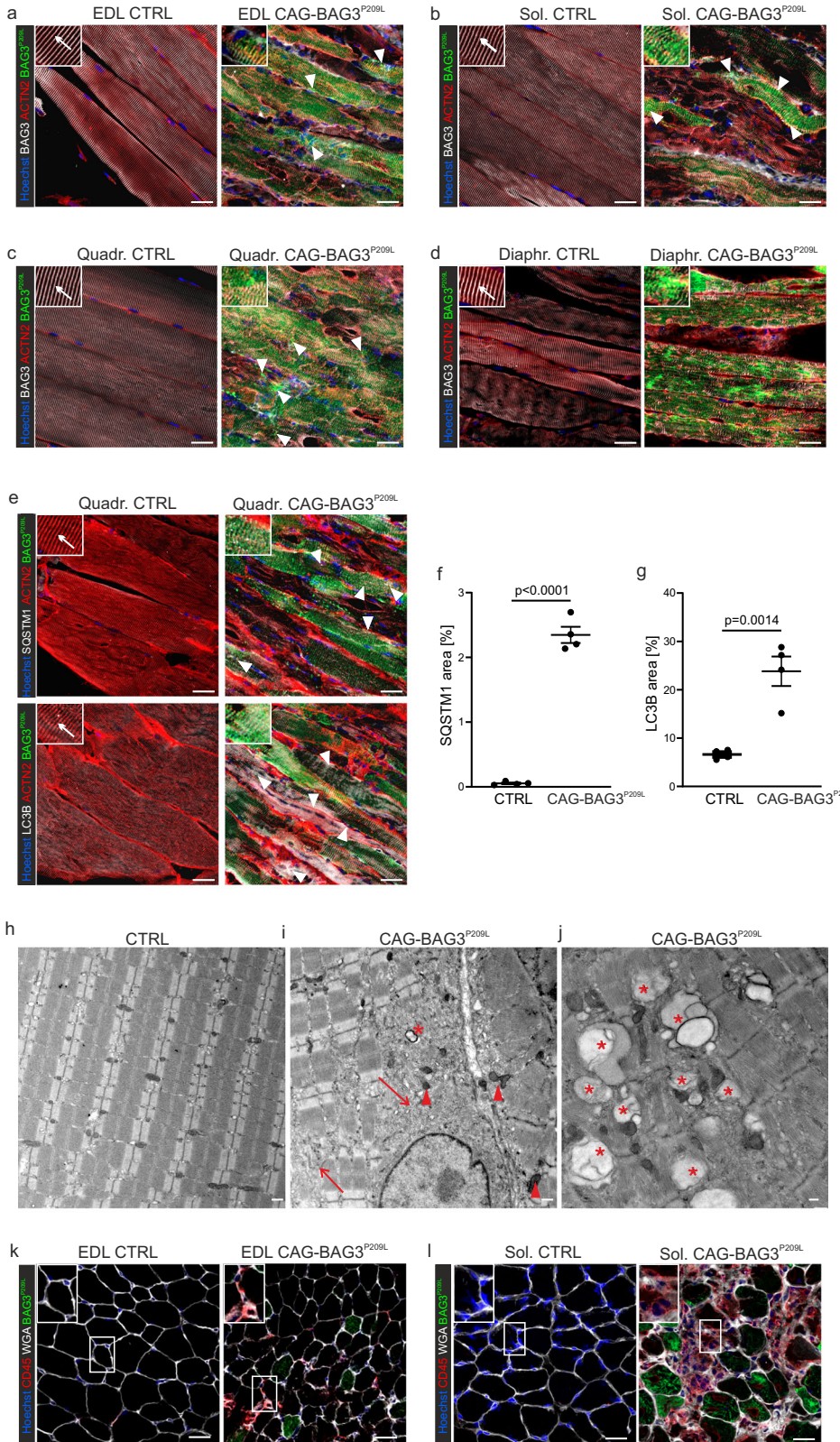

## Massive protein accumulation and damaged mitochondria in CAG-BAG3^P209L-mice

We took an unbiased approach and performed a multi-omics analysis of the quadriceps femoris muscle from 5- week-old CAG-BAG3^P209L-mice and their matching controls. RNA-Seq analysis, in which a total of 4 CAG-BAG3^P209L mice were compared with 5 control mice, revealed 1981 significantly differentially expressed genes, of which 1105 were upregulated and 876 were downregulated (Fig. 4a, Supplementary Tables 1+2). GO-analysis revealed a strong expression of genes associated to protein synthesis and translation in skeletal muscle from CAG-BAG3^P209L-mice (Fig. 4b). As mTOR-signaling was among the enriched terms, we hypothesized that this pathway caused the increase in protein synthesis genes. Increased transcription by RNA polymerase II was also evident from the GO-analysis. Conversely,

**Fig. 3 | Impairment of sarcomere structure in skeletal muscle from 4 weeks old CAG-BAG3[P209L]-mice. a–d** Tissue of the EDL and soleus muscle of controls (CTRL) showed distinct cross-striation (arrows in inserts) after staining for BAG3, ACTN2 and DES, whereas aggregates of all three proteins were detected in the muscle tissue of CAG-BAG3[P209L]-mice (**a–d**, arrowheads). **e** Staining for SQSTM1 (white), LC3B (white), ACTN2 (red) and BAG3 (green) in quadriceps femoris muscle from CTRL and CAG-BAG3[P209L] mice, **f, g** Quantification of SQSTM1 (**f**) and LC3B (**g**) stained area in muscle cells from CTRL and CAG-BAG3[P209L] quadriceps muscle. *n* = 4 biologically independent mice per group; shown is mean ± SEM, statistical test = two-sided unpaired Student's *t*-test. **h–j** Electron micrographs obtained from quadriceps femoris muscle of control and CAG-BAG3[P209L]-mice. **h** Skeletal muscle of controls with regular arrangement of sarcomeres and typical transverse striation. **i, j** Skeletal muscle from CAG-BAG3[P209L]-mice with disintegration of sarcomeres and intersarcomeric electron dense regions (**i**, arrows), disarranged mitochondria displaying a variety of shapes and sizes (**i**, arrowheads) and formation of autophagic vacuoles with signs of mitophagy (**i, j**, asterisks). **k, l** Staining for CD45-positive cells in EDL and soleus muscle of CTRL and CAG-BAG3[P209L]-mice. Scale bars = 25 µm (**a–e, k, l**), 500 nm (**h**), and 250 nm (**i, j**). **a–l** Data are representative of three experiments with biologically independent replicates. Source data are provided as a Source Data file.

expression of genes involved in catabolic processes such as autophagy and macroautophagy was upregulated, too. Moreover, protein degradation terms such as "ubiquitin-dependent protein catabolic process", "proteasome-mediated ubiquitin-dependent protein catabolic process", and "endoplasmic-reticulum-associated protein degradation pathway" were among the enriched GO-terms. Interestingly, hypoxia response gene expression was also upregulated, which could indicate reduced blood flow and thus oxygen supply due to impaired cardiac output in these mice[6]. These data demonstrated that increased protein synthesis is the response of a highly adaptable tissue to sarcomeric damage caused by impaired autophagy due to the expression of BAG3[P209L].

In addition, we found the GO-terms "response to mitochondrial depolarization", "autophagy of mitochondrion", and "mitochondrion disassembly", indicating mitochondrial alterations among the genes that were at least 2-fold upregulated ($\log_2$ fold change ≥ 1) (Fig. 4b). This was consistent with the ultrastructural analysis, which revealed irregular shapes of mitochondria in skeletal muscle of CAG-BAG3[P209L]-mice when compared to controls (Fig. 3g–i).

Among the GO-terms associated with downregulated gene expression were "mitotic cell cycle" and "muscle cell proliferation", including genes such as *AurkB* and several cyclins. A detailed analysis of the downregulated GO-term "system development" revealed that "vasculature development" and "endothelium development" were affected. Among the other downregulated GO-terms was a strong signature of angiogenesis genes and "cell migration". Together this could indicate a decrease in capillarization, which in turn could lead to hypoxia. In fact, the capillary density in the soleus muscle was significantly decreased in CAG-BAG3[P209L] mice (Supplementary Fig. 5h). Interestingly, neurogenesis and neuronal genes were also among the depleted GO-terms, hinting at a neuronal phenotype that has been described in patients suffering from MFM6[2]. We also observed decreased fatty acid metabolism in skeletal muscle, manifested by depletion of the GO category "fatty acid beta-oxidation" (Fig. 4b), pointing to impaired mitochondrial function.

For proteome analysis, the quadriceps femoris muscle from 5-6 week-old CAG-BAG3[P209L]-mice was dissected and total protein extracted. Label-free mass spectrometry analysis in data-independent acquisition (DIA) mode identified 3399 protein groups in the total lysate. 971 protein groups significantly increased in abundance (*t*-test with permutation-based FDR < 0.05, S0 = 0.1) and 68 protein groups significantly decreased in abundance in skeletal muscle of the CAG-BAG3[P209L]-mice when compared to the controls (Fig. 5a, Supplementary Table 3). In addition, 215 protein groups were found exclusively in the CAG-BAG3[P209L] mouse muscle lysate, and 4 protein groups in the control lysate (Supplementary Table 4). GO-CC term enrichment analysis of the 1186 protein groups with exclusive or higher abundance in mutant mice showed that these were enriched in proteins associated with the sarcomere, myofibril, and actin cytoskeleton (Supplementary Fig. 6a). Analysis of KEGG pathways further revealed that accumulating proteins were involved in protein synthesis (ribosome, spliceosome, protein processing in the ER), protein degradation (proteasome, phagosome, lysosome, mitophagy), as well as components of mechanosensitive structures (focal adhesions, regulation of actin

cytoskeleton, adherens junctions) (Supplementary Fig. 6b). Similar to our previous analysis of heart samples from CAG-BAG3[P209L]-mice[6], components of the CASA-complex such as BAG3, HSPB8, and other heat shock proteins, such as CRYAB, HSPB1, DNAJB2, and DNAJB5 massively accumulated in CAG-BAG3[P209L]-mice. Thus, expression of BAG3[P209L] massively perturbs skeletal muscle proteostasis, leading to the accumulation of a large but specific set of proteins in 5–6 week-old mice. This was not observed in skeletal muscle from CAG-BAG3[WT]-mice (Suppl. Fig. 7).

To verify the changes in the protein quality control system observed in proteomics analysis, we looked at protein expression focusing on alterations in heat shock protein expression, CASA, and autophagy. The quadriceps femoris muscle was dissected from 5 weeks old CAG-BAG3[P209L], CAG-BAG3[WT] and control mice and protein extracts were prepared for Western Blot analysis (Fig. 5b, Supplementary Fig. 7a). In agreement with the mass spectrometry data, we observed a statistically significant increase in BAG3, CRYAB (Fig. 5c, d) and the small heat shock proteins HSPB7 and 8 (Fig. 5e, f). Expression of HSPB6 showed a consistent trend of increase that did not reach statistical significance due to greater variations in protein abundance (Fig. 5g). The autophagy markers LC3B and SQSTM1, and the CASA-related protein SYNPO2 were significantly increased in muscles from CAG-BAG3[P209L]-mice (Fig. 5h–j). These protein accumulations, which did not take place in CAG-BAG3[WT]-mice (Supplementary Fig. 7f–l), suggest cellular stress and altered autophagy, in agreement with the ultrastructural data that showed an increase in autophagic vesicles (Fig. 3i). Together, this indicated a dysregulation of autophagy and a reduced ability to clear non-soluble protein accumulations.

## Subcellular fractionation reveals massive protein aggregation with a shift from the soluble to the insoluble fraction

Since the dysregulation of autophagy led to the formation of protein aggregates, we analyzed these in more detail. To distinguish between aggregating proteins and proteins that might accumulate in soluble form as a stress response, we further separated protein extracts from skeletal muscle into a soluble and a detergent-resistant insoluble fraction (Fig. 6a). Label-free DIA mass spectrometry identified 3838 protein groups in the soluble and 2539 in the urea-dissolved pellet fractions. Comparison of pellet and soluble fractions showed the expected enrichment of cytoskeletal proteins in the pellet in both wild-type and CAG-BAG3[P209L]-expressing mice (Supplementary Fig. 8a–c), thereby confirming the method. Compared to the total cell lysate, analysis of the separated fractions identified 1059 additional protein groups, while 1975 of the protein groups were observed in all three proteomes (lysate/soluble/insoluble, Supplementary Fig. 8d). To determine which proteins accumulate in the observed protein aggregates, we compared protein abundance in the insoluble fraction between the two strains. Using the same significance criteria as above (*t*-test with permutation-based FDR < 0.05) revealed that 961 protein groups increased in abundance in the pellet of CAG-BAG3[P209L]-mice (Fig. 6b, Supplementary Table 5), and an additional 500 proteins were only observed in the insoluble pellet fraction of the CAG-BAG3[P209L]-mice but not in the controls (Supplementary Table 6). KEGG term analysis of these 1461 protein groups showed the enrichment of

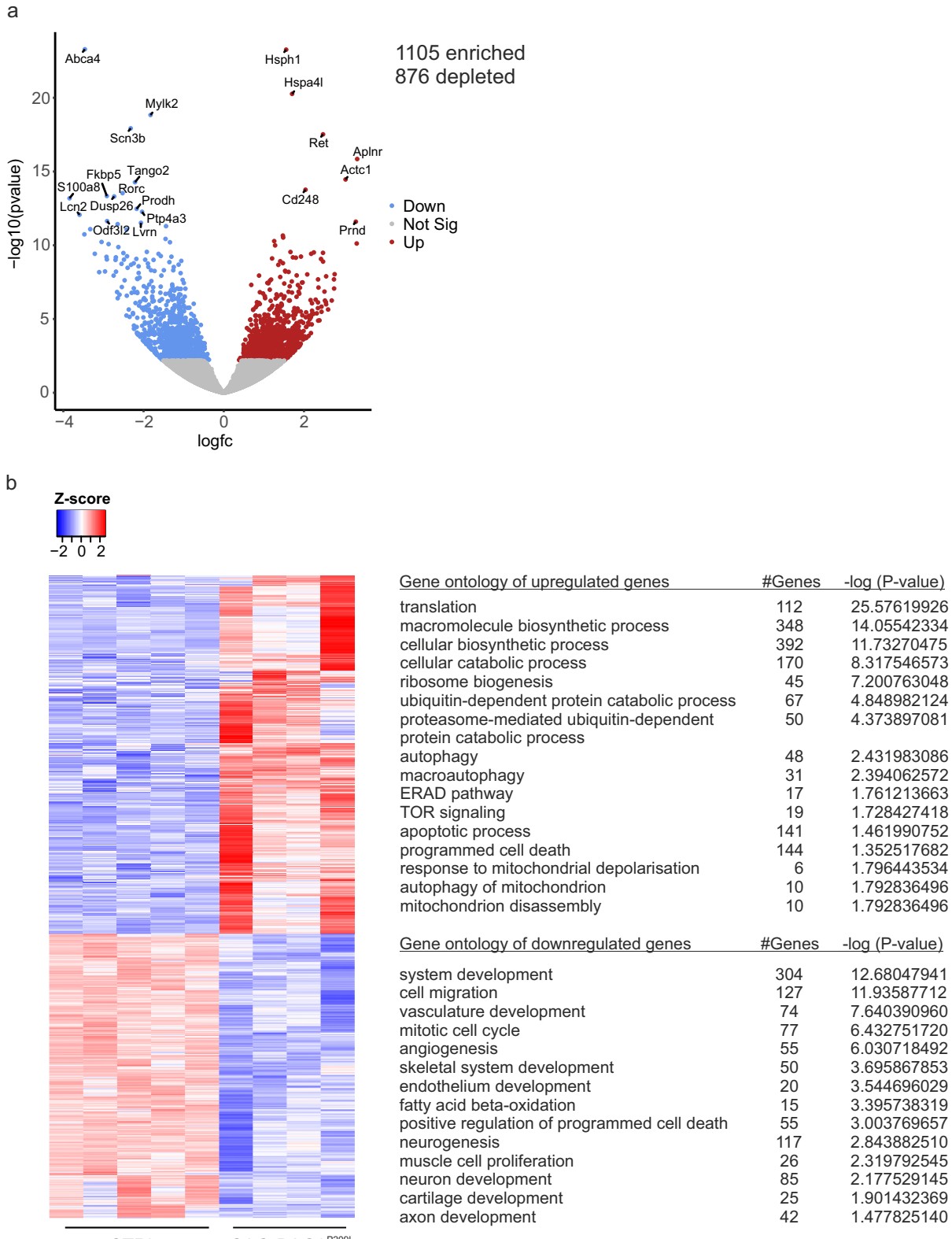

**Fig. 4 | RNA-Seq-based transcriptome analysis of skeletal muscle from 5 weeks old CAG-BAG3P209L and control mice. a, b** RNA-Seq analysis revealed differential gene expression in skeletal muscle from 5-week-old CAG-BAG3P209L-mice ($n = 4$ biologically independent mice) compared to controls ($n = 5$ biologically independent mice). **a** Volcano plot showing false discovery corrected $q$-values and log$_2$(fold)-changes. Blue and red dots depict genes depleted and enriched respectively in CAG-BAG3P209L-mice compared to controls. Names of the the 20 most differentially expressed genes are shown. Statistical test = two-sided Wald-test. **b** Same skeletal muscles as in (**a**). Gene expression heatmap of all significantly differentially expressed genes and list of significantly enriched and depleted GO-terms (−log (FDR value)). Source data are provided as a Source Data file.

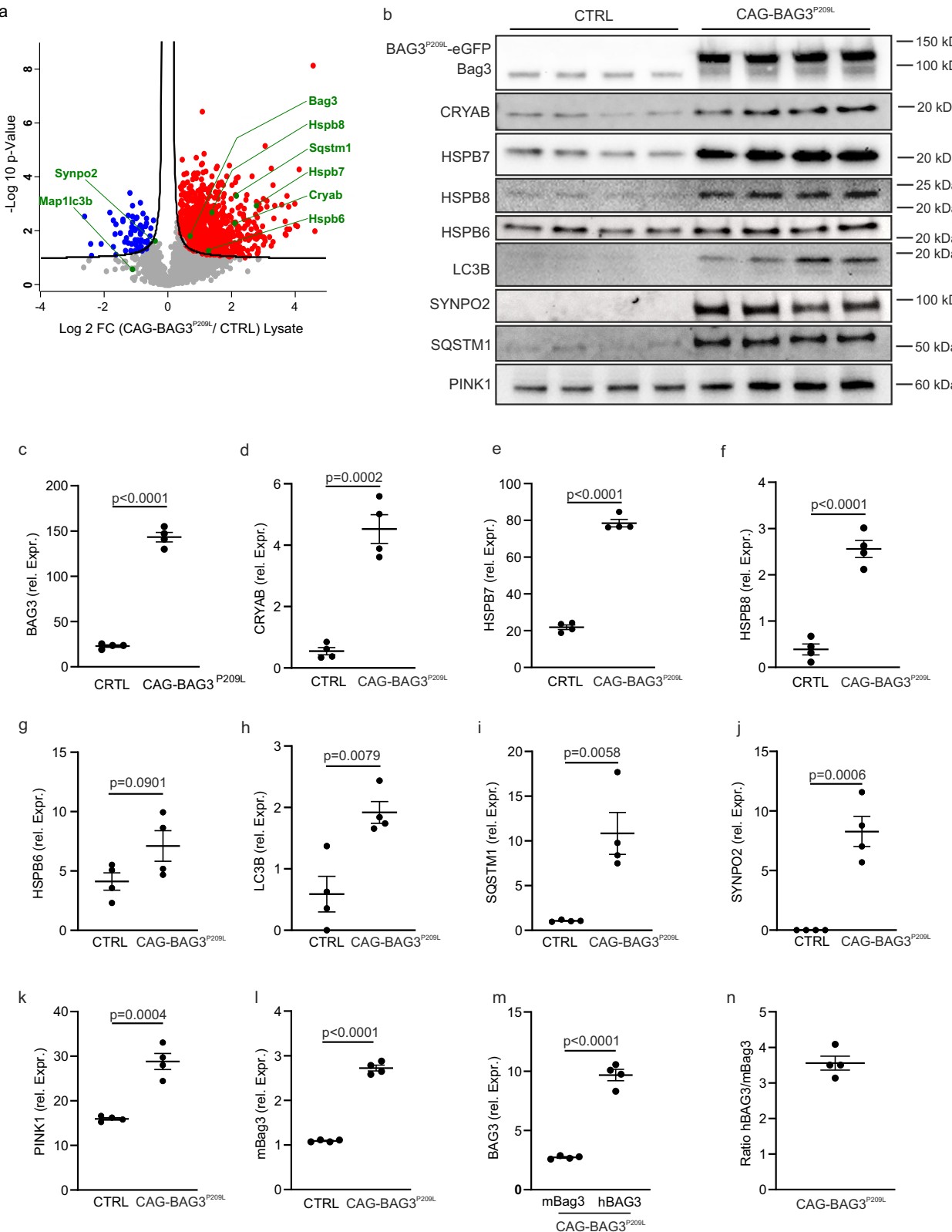

proteins involved in autophagy, mitophagy, mTOR signaling, and mitochondrial processes such as oxidative phosphorylation (Suppl. Fig. 9a). In contrast, only 30 proteins showed decreased abundance in the pellet from CAG-BAG3[P209L] mice (Supplementary Table 6). In the soluble fraction, 616 protein groups accumulated, while 248 protein groups decreased in abundance in the CAG-BAG3[P209L]-mice (Fig. 6c, Supplementary Table 7), with 46 protein groups observed exclusively

in CAG-BAG3[P209L] and 3 protein groups in the control mice (Supplementary Table 8). Notably, proteins accumulating in the soluble fraction were mainly associated with protein turnover, including the ribosome, spliceosome, and proteasome (Suppl. Fig. 9b). Comparing the change in protein abundance between the two strains in both pellet and supernatant showed a weak positive correlation (Pearson's correlation Coefficient r = 0.2, −log10 (Pearson p-value) = 13.9) but also

**Fig. 5 | Accumulation of heat shock proteins and autophagy markers in skeletal muscle of CAG-BAG3^P209L^-mice. a** Quantitative proteome analysis of proteins identified in the total lysate of control and CAG-BAG3^P209L^-mice. Proteins significantly (two-sided unpaired Student's *t*-test with an FDR corrected *p*-value ≤ 0.05) are highlighted in red (accumulating in the CAG-BAG3^P209L^-mice) and blue (depleted in CAG-BAG3^P209L^-mice). **b** Western blot analyses were performed on quadriceps femoris muscle of controls and CAG-BAG3^P209L^-mice at 4 weeks of age. **c**–**m** Quantification of **c** total BAG3 (mouse Bag3 and human BAG3^P209L^), **d** heat

shock proteins CRYAB, **e** HSPB7, **f** HSPB8, **g** HSPB6, **h** autophagy markers LC3B and **i** SQSTM1, **j** the BAG3 binding partner SYNPO2, **k** the mitophagy marker PINK1, and **l** endogenous mouse Bag3. **m** Quantification of mBag3 and hBAG3 in skeletal muscle from CAG-BAG3^P209L^-mice. **n** Ratio of hBAG3 to mBag3 in skeletal muscle from CAG-BAG3^P209L^-mice. *n* = 4 biologically independent mice per group, shown is the mean ± SEM, statistical test = two-sided unpaired Student's *t*-test, rel. Expr. = Expression relative to total protein. Source data are provided as a Source Data file.

### Table 1 | Respiratory parameters of isolated skeletal muscle mitochondria

| | | Glut-mal | ADP-250 µM | state 4 | RCI | Creatine | Crea-effect | CAT | CAT-RCI | TTFB |
|---|---|---|---|---|---|---|---|---|---|---|
| CTRL | MW | 22.59 | 292.3 | 51.59 | 5.76 | 222.8 | 3.026 | 22.86 | 11.45 | 236.2 |
| *n* = 3 (6) | SD | 3.40 | 62.89 | 6.7 | 1.5 | 27.85 | 0.18 | 12.7 | 4.32 | 66.6 |
| CAG-BAG3^P209L^ | MW | 16.3 | 186.5 | 56.9 | 3.3 | 150.7 | 1.76 | 16.62 | 9.49 | 174.6 |
| *n* = 3 (6) | SD | 7.05 | 16.03 | 8.72 | 0.697 | 31.1 | 0.11 | 3.78 | 2.94 | 37.76 |
| | *p* | 0.045 | 0.092 | 0.452 | 0.098 | 0.004 | 4.6E−06 | 0.344 | 0.428 | 0.043 |
| | | OC-mal | ADP-250 µM | State 4 | RCI | Creatine | Crea-effect | CAT | CAT-RCI | TTFB |
| CTRL | MW | 24.05 | 87.30 | 49.91 | 1.76 | 89.22 | 1.20 | 26.96 | 3.32 | 89.86 |
| *n* = 3 (6) | SD | 5.48 | 0.54 | 6.26 | 0.22 | 11.11 | 0.08 | 3.34 | 0.420 | 7.00 |
| CAG-BAG3^P209L^ | MW | 18.84 | 61.89 | 55.21 | 1.11 | 56.85 | 0.94 | 19.39 | 2.91 | 55.82 |
| *n* = 3 (6) | SD | 6.56 | 14.81 | 6.99 | 0.18 | 19.66 | 0.049 | 5.61 | 0.25 | 16.75 |
| | *p* | 0.16 | 0.096 | 0.38 | 0.018 | 0.084 | 0.017 | 0.131 | 0.228 | 0.002 |

glut-mal—substrate combination 10 mM glutamate + 5 mM malate, OC-mal—substrate combination 1 mM octanoylcarnitine + 5 mM malate, CAT—1 µM carboxyatractyloside, TTFB—-1 µM umcoupler, RCI respiratory control index. The experiments were performed in duplicates with 3 biologically independent replicates. The respiratory rates are expressed in nmol O$_2$/min/mg protein. Shown is the mean ± SD, statistical test = two-sided unpaired Student's *t*-test. Source data are provided as a Source Data file.

revealed that most proteins accumulated significantly more in the insoluble fraction (Fig. 6d).

GO-CC term enrichment analysis of proteins depleted in both soluble and pellet fractions indicated that this group included many sarcoplasmic and sarcomeric proteins, in line with the observed phenotype (Fig. 6e). Notably, 215 proteins accumulating in the pellet fraction (log$_2$(BAG3^P209L^/control) > 0.25) showed decreased abundance in the soluble fraction (log$_2$ (BAG3^P209L^/control) < −0.25), indicating increased association with the cytoskeleton or aggregation (Fig. 6d). GO-CC term analysis indicated that many of these proteins were of mitochondrial origin (Fig. 6f). Since BAG3 has previously been implicated in regulating mitophagy[10], we wondered if this accumulation of mitochondrial proteins in the pellet might coincide with mitophagy markers. Indeed, we observed a strong accumulation of the PARKIN target MFN1 and the mitophagy-regulating Bcl-2 family members BNIP3 and BCL2L13 in the pellet of CAG-BAG3^P209L^ mice (Fig. 6b), but not in the supernatant (Fig. 6c), while citrate synthase (CS), a marker of total mitochondrial protein, showed no significant change between the two genotypes. In addition, the mitophagy receptor PINK1, and several other factors involved in mitophagy (ATF5, ULK1, MFN2, SIRT4, KEAP1) were found only in the pellet fraction of CAG-BAG3^P209L^ mice, but not in controls (Fig. 6G, Supplementary Table 6). In summary, the omics-analysis revealed impaired autophagy and mitophagy, both of which could be responsible for the manifestation of the pathological phenotype.

### Functionally impaired mitochondria in skeletal muscles from CAG-BAG3^P209L^-mice

Proteomic analysis revealed mitochondrial proteins and mitophagy markers as components of the aggregates observed in both immunofluorescence and ultrastructural analysis. The latter illustrated that most mitochondria displayed an irregular shape, were disorganized, and showed signs of mitophagy in skeletal myocytes from CAG-BAG3^P209L^-mice. These data suggested that mitochondria could play a role in the pathomechanism of the disease, and therefore, we analyzed their biochemical properties and their ability to provide energy. To this end, we isolated intact mitochondria from total mouse hindlimb

### Table 2 | Enzyme pattern of skeletal muscle mitochondria

| | | CS/ protein | COX/ protein | COX/ CS | Complex I/protein | Complex I/CS |
|---|---|---|---|---|---|---|
| CTRL | MW | 2.69 | 21.39 | 7.95 | 0.75 | 0.26 |
| *n* = 3 (6) | SD | 0.14 | 2.00 | 0.86 | 0.11 | 0.04 |
| CAG-BAG3^P209L^ | MW | 1.96 | 15.38 | 7.94 | 0.39 | 0.18 |
| *n* = 3 (6) | SD | 0.55 | 3.63 | 1.03 | 0.12 | 0.01 |
| | *p* | 0.021 | 0.007 | 0.98 | 0.005 | 0.005 |

*CS* citrate synthase, *COX* cytochrome c oxidase, *complex I* rotenone-sensitive NADH:CoQ1 oxidoreductase. The experiments were performed in duplicates with 3 biologically independent replicates. The activities are expressed in units/mg protein. Shown is the mean ± SD, statistical test = two-sided unpaired Student's *t*-test. Source data are provided as a Source Data file.

muscles. As presented in Table 1, oxygraph measurements revealed that the maximum respiration capacity of isolated skeletal muscle mitochondria obtained by uncoupling is significantly lower for both substrate combinations glutamate + malate, which measures respiration without fatty acids and octanoylcarnitine and malate, which examines fatty acid oxidation. Resting state respiration activities (state 4) remained unaltered. Additionally, we observed that the stimulatory effect of creatine on mitochondrial respiration was greatly reduced for both substrate combinations (Table 1). To elucidate the cause of the observed functional alterations of skeletal muscle mitochondria, we further analyzed the enzyme pattern of isolated mitochondria (Table 2). We observed that the activities of all enzymes expressed per protein were significantly decreased. Importantly, the ratio to the marker enzyme citrate synthase was significantly lower for complex I, while the cytochrome c oxidase/citrate synthase ratio was unaltered, suggesting additionally a specific impairment effect of transgene expression on complex I activity and ruling out potential effects of the purification procedure. Since complex I is required for glutamate +malate and also octanoylcarnitine+malate respiration, its functional impairment explains the remarkable reduction in the maximal

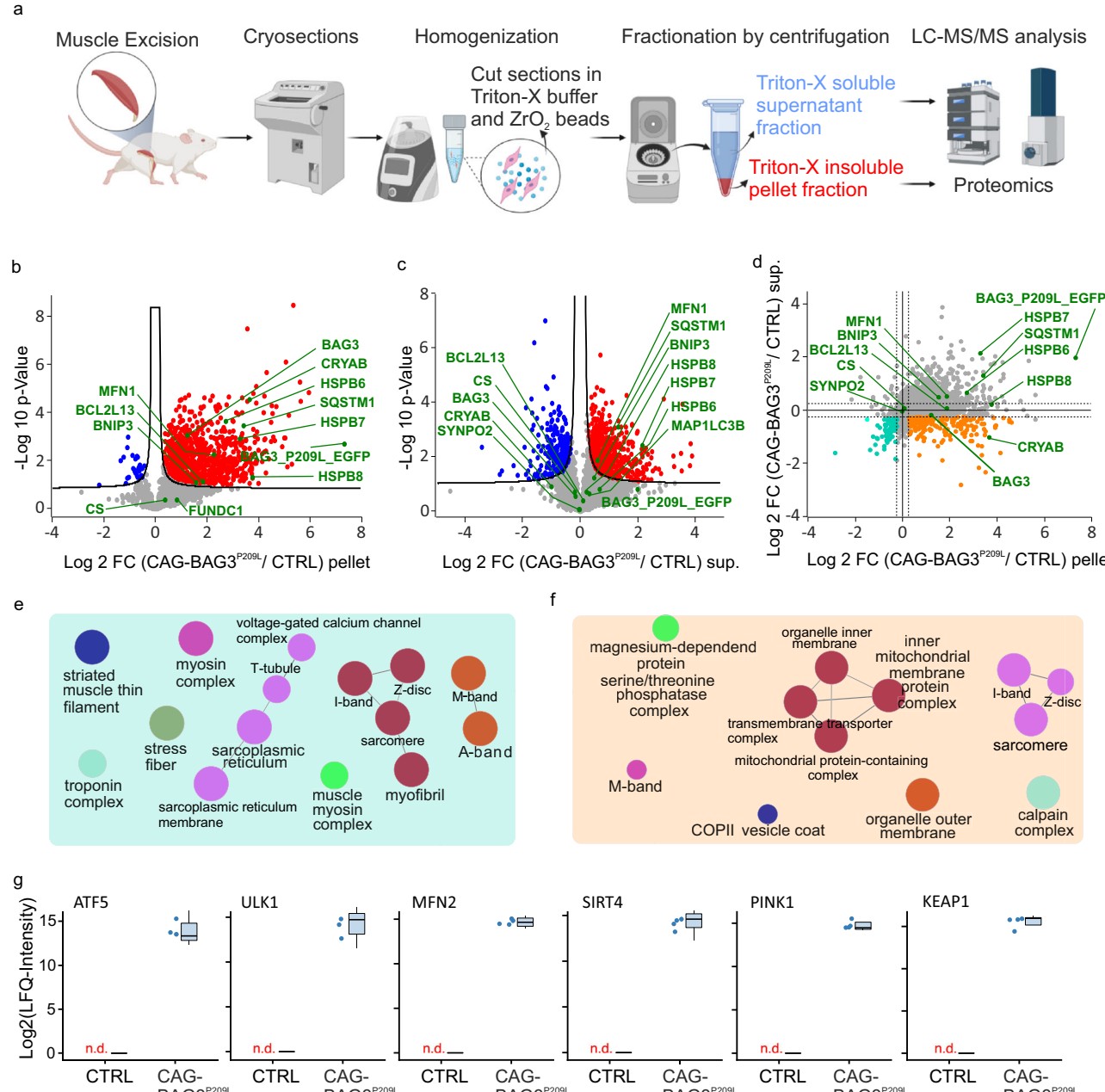

**Fig. 6 | Quantitative analysis of fractionated skeletal muscle. a** Schematic of the workflow. Frozen mice quadriceps muscles were cut into thin sections using a cryotome, and homogenized in Triton-X 100 buffer. Proteins that were not soluble in the Triton-X buffer were centrifuged down and solubilized using Urea buffer. The soluble supernatant and the insoluble pellet were trypsin digested and analyzed using a liquid chromatography setup coupled to a tandem mass spectrometer. Created in BioRender. Kuppusamy, M. (2026) https://BioRender.com/6mj1kxu. $n = 4$ biologically independent mice per group. **b**, **c** Volcano plots show significant differences in protein abundance between control and hBAG3[P209L]-expressing mice in **b** the pellet fraction and **c** the supernatant fraction. Proteins with significant differences (Student's $t$-test with a permutation-based FDR corrected $p$-value $\leq$ 0.05) in abundance are highlighted in red (accumulating in the CAG-BAG3[P209L]-mice) and blue (depleted in CAG-BAG3[P209L]-mice). $n = 4$ biologically independent mice per group. **d** Comparison of the difference in protein abundance between the pellet (determined in panel (**b**), plotted on $x$-axis) and the supernatant (determined in panel (**c**), plotted on $y$-axis). Proteins accumulating in the CAG-BAG3[P209L] skeletal muscle but depleted in the CAG-BAG3[P209L] heart muscle are highlighted in orange and proteins depleted in both the soluble and the pellet fraction are highlighted in green. **e**, **f** Gene ontology cellular component (GO-CC) term enrichment for **e** proteins depleted in both the supernatant and the pellet, and **f** for proteins accumulating in the pellet but depleted in the supernatant. Only terms with a $p$-value $\leq 0.05$ are shown. **g** Quantification of selected proteins identified only in the pellet fraction of BAG3[P209L]-expressing mice, but not in controls (n.d.). $n = 3$ biologically independent mice per group for ATF5 and ULK1; $n = 4$ biologically independent mice per group for MFN2, SIRT4, PINK1, and KEAP1. Box plots show the median (center line), the 25th and 75th percentiles (box), and the minimum and maximum values (whiskers). Source data are provided as a Source Data file.

respiration activities detected in the oxygraphic determinations (Table 1).

In contrast to the decreased enzymatic activities of isolated mitochondria in whole skeletal muscle, mitochondrial accumulation was evident in COX-SDH staining, which was markedly increased in the quadriceps muscle of CAG-BAG3[P209L]-mice. (Suppl. Fig. 10). This is likely due to an accumulation of damaged and functionally impaired mitochondria as seen in electron micrographs (Fig. 3i), since

oxygraphic and enzymatic data of isolated skeletal muscle mitochondria show considerably decreased activities. Most likely, these effects can be attributed to an impairment of mitophagy caused by BAG3[P209L] overexpression. This is underlined by the upregulation of *Pink1*, *Bnip3l*, and *Tomm20* seen in the RNA-seq analysis (Supplementary Table 1). Moreover, in the proteomics analysis PINK1, BNIP3, and FUNDC1 were enriched in the detergent-resistant insoluble fraction (Supplementary Tables 5 + 6), while TOMM22, TOMM40, and TOMM70 were enriched in the soluble fraction (Supplementary Table 7). The observed enrichment of proteins functioning as different mitophagy receptors, like PINK1, BNIP3, and FUNDC1, strongly indicates increased activation of different mitophagy pathways despite insufficient final removal of functionally impaired mitochondria. This can be explained by a block of final execution of mitophagy due to the loss of BAG3 function.

### Block in autophagy is the primary cause of the phenotype, while impairment of mitochondria is secondary

Next, we asked whether the observed severe skeletal muscle phenotype was mainly caused by impaired autophagy, mitophagy, or both. To test this, we isolated skeletal myocyte precursor cells from the skeletal muscles of CAG-BAG3[P209L] mice using an established protocol[11]. After differentiation into skeletal myocytes, these cells displayed formation of BAG3[P209L]-positive protein aggregates (Fig. 7a) and could therefore be used to study the effects of inducing autophagy, mitophagy, or mitochondrial biogenesis. We transfected these cells with plasmids carrying expression cassettes for *Becn1* to induce general autophagy, *Bnip3* to specifically induce mitophagy, or *Ppargc1a* to induce mitochondrial biogenesis. Successful overexpression of these proteins was confirmed by positive staining for the Myc (BECN1), FLAG (BNIP3), or HA (PPARGC1a) tags fused to the different proteins (Fig. 7a).

Since bafilomycin A1 blocks the fusion of autophagosomes with lysosomes, an increase in autophagic markers like SQSTM1 indicates the level of autophagic flux, while an increase in mitophagy markers such as PINK1 reflects the level of mitophagic flux. There was a statistically significant increase in SQSTM1 in the BECN1-overexpressing cells, which was consistent with a rise in autophagic flux and the release of the autophagic block seen in cells from CAG-BAG3[P209L] mice (Fig. 7b). This was not observed after induction of mitophagy (BNIP3) or increased mitochondrial biogenesis (PPARGC1a). PINK1 accumulated in BNIP3-overexpressing cells after Bafilomycin A1 treatment, confirming that the induction of mitophagy was successful (Fig. 7c). Interestingly, mitophagy was also induced by BECN1 overexpression, as PINK1 levels decreased (Fig. 7c), which had initially accumulated due to a block in mitophagy. As expected, the level of BAG3[P209L]-eGFP increased following BECN1 overexpression and Bafilomycin A1 treatment, indicating enhanced autophagic flux and degradation of BAG3[P209L]-eGFP-positive protein aggregates (Fig. 7d). In contrast, neither the induction of mitophagy through overexpression of BNIP3 nor the promotion of mitochondrial biogenesis via PPARGC1a overexpression affected autophagy, mitophagy, nor the amount of BAG3-eGFP containing protein aggregates (Fig. 7b–e). In addition, cell morphology improved after overexpressing BECN1, as the average size of skeletal myocytes was larger (Fig. 7a) and had a more prominent cross-striation. Unbiased analysis of sarcomere quality using the SarcAsM algorithm[12] revealed a statistically significant increase in Z-band length in BECN1 transfected cells compared to CTRL and PPARGC1a transfected cells (Suppl. Fig. 11a). Furthermore, cells transfected with BECN1 showed a tendency toward increased sarcomere length, sarcomere area ratio, and cell area ratio (Suppl. Fig. 11b-d), supporting the morphological observations.

We expanded these experiments in vivo by inducing autophagy in 3-week-old CAG-BAG3[P209L] mice through daily i.p. injection of 8 mg/kg rapamycin for 14 days. Rapamycin is known to induce autophagy by inhibiting the mTOR signaling pathway[13]. Grip strength and motor coordination, measured by the Rota-Rod device, improved on day 7 and day 14 under rapamycin treatment compared to untreated or vehicle-treated mice (Suppl. Fig. 11e, f). The same was true for voluntary movements (Suppl. Fig. 11g). The amount of LC3BII protein was significantly increased in the quadriceps muscles of rapamycin-treated CAG-BAG3[P209L] mice after 14 days of rapamycin treatment (Suppl. Fig. 11h), indicating enhanced autophagy. Together with the in vitro data, we conclude that the phenotype in the skeletal muscle of CAG-BAG[P209L] mice is caused by a blockage of autophagy and that the impairment and accumulation of mitochondria are secondary to this.

### Improved skeletal muscle strength after AAV-mediated knockdown of hBAG3[P209L]

Gene therapy offers a promising approach for treating monogenetic inherited diseases. As an experimental approach, we developed a knockdown strategy targeting the human BAG3[P209L]-eGFP fusion protein. The construct consisted of an expression cassette with an U6 promoter driving the expression of either a *hBAG3*-specific shRNA or a scrambled shRNA along with a CMV-promoter for mCherry expression as an indicator of successful transduction. For delivery of the shRNA expression construct, we utilized an adeno-associated virus (AAV), which was injected into the jugular vein of P15 mice. The transduction rate was determined by counting the mCherry-positive skeletal muscle myocytes, while the effect on BAG3[P209L]-eGFP expression was monitored by measuring the total eGFP intensity in sections from skeletal muscles (Suppl. Fig. 12a, b). Both AAVs displayed a high transduction rate of approximately 80% (Supplementary Fig. 7b), but only the shRNA AAV was able to visibly reduce BAG3[P209L]-eGFP expression (Fig. 8a–c). Additionally, the number of BAG3[P209L]-eGFP containing protein aggregates was reduced, and a regular sarcomere structure was obvious after staining for ACTN2, DES, and FLNC (Suppl. Fig. 12c–e). Western Blot analysis confirmed the immunofluorescence findings (Suppl. Fig. 13a-n) and showed a reduction in BAG3 expression (by ~60%, Fig. 8c) and autophagy markers (Suppl. Fig. 13i, m). Centralization of nuclei in skeletal muscles decreased significantly in the shRNA-treated group in the quadriceps muscle (from $35.9 \pm 4.1$ to $7.0 \pm 3.8\%$ of fibers) together with a reduction in granular areas (Fig.8d, e).

Fibrosis was markedly reduced after BAG3 shRNA-treatment (from $23.8 \pm 3.7$ to $6.8 \pm 0.8\%$ of fibers in quadriceps muscle, Fig. 8f, g). Most strikingly, force measurements following AAV treatment revealed a significant increase in normalized force development in both soleus and EDL muscles (Fig. 8h, i), demonstrating a functional improvement (EDL muscle from 4 to 102 mN/mm$^2$ and soleus muscle from 53 to 301 mN/mm$^2$ at 100 Hz). Also, both the soleus and EDL muscles exhibited a significant improvement in muscle length and weight (Suppl. Fig. 13n–q). In summary, we demonstrate a significant improvement in the pathological phenotype using this proof-of-concept gene therapy approach for myofibrillar myopathies.

## Discussion

Myofibrillar myopathies are diseases characterized by protein aggregation that lead to sarcomere disintegration, causing muscle weakness and cardiomyopathies. The most severe form of MFM known so far is MFM6, caused by the P209L-mutation in the co-chaperone BAG3. Accordingly, in the skeletal muscle of CAG-BAG3[P209L]-mice, we discovered the typical hallmarks of myofibrillar myopathies, such as the formation of cytoplasmic protein inclusions and disintegration of sarcomere structures. This was caused by impaired and blocked autophagy, followed by a secondary blockage in mitophagy.

BAG3 plays a crucial role in protein homeostasis by clearing damaged proteins through autophagy[1,14] and by regulating translation through the spatial organization of the mTORC1 complex[15]. Introduction of the P209L-mutation into the mouse Bag3 (P215L) did not lead to an obvious phenotype[4], most likely caused by amino acid sequence

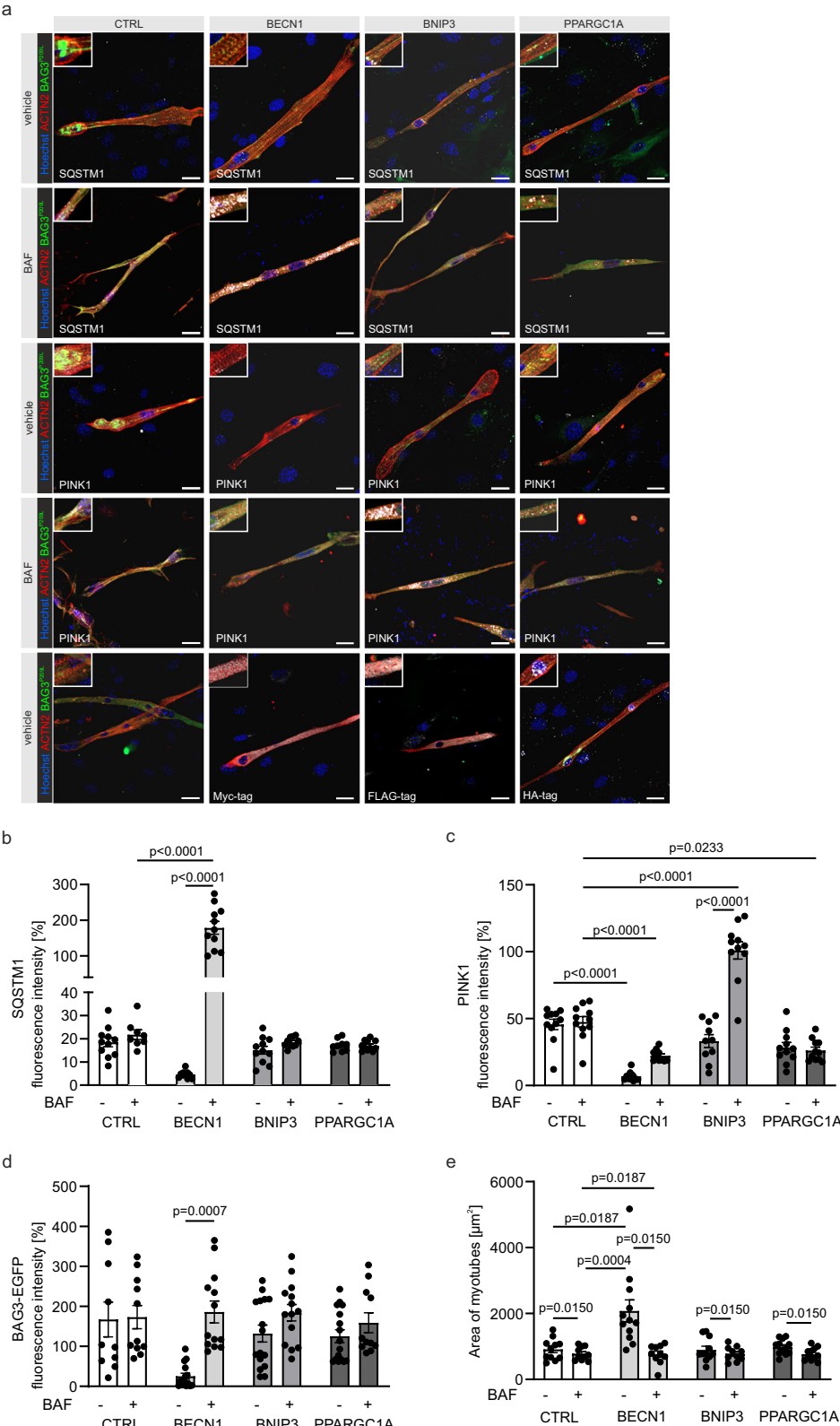

variations in regions that are not conserved compared to human BAG3 (Suppl. Fig. 14). In contrast, overexpression of human BAG3^P209L-eGFP in mice impairs BAG3 function and leads to phenotypes very similar to those observed in MFM6 patients. This was not due to an unspecific effect of overexpressing a large amount of the BAG3^P209L-eGFP fusion protein, as a CAG-BAG3^WT control mouse line did not show a patho-logical phenotype, despite expressing comparable amounts of *BAG3^WT*-

*eGFP* mRNA (total mRNA ~3-times higher than in non-transgenic CTRLs). However, the amount of BAG3 protein in CAG-BAG3^WT was only ~4-times higher than in CTRLs, whereas it was ~11 times higher in CAG-BAG3^P209L mice, consistent with the accumulation of BAG3^P209L-eGFP protein that is part of the pathological phenotype.

CAG-BAG3^P209L-mice displayed growth restriction and muscular atrophy, starting at 2 weeks of age, which worsened with age[6]. This

**Fig. 7 | The phenotype in skeletal muscle of CAG-BAG^P209L mice is due to a blockage of autophagy. a** Primary myoblasts from CAG-BAG3^P209L-mice were differentiated into skeletal myocytes and transfected with BECN1, BNIP3, PPARGC1A, or vehicle (CTRL) (top row). Cells were treated with vehicle or bafilomycin A1 (BAF) as indicated. Immunostaining was performed for SQSTM1 (white), PINK1 (white), and ACTA2 (red). Images are representative of independent myoblast differentiations derived from biologically independent mice. EGFP (green) marks BAG3^P209L-eGFP overexpression; nuclei were counterstained with Hoechst (blue). Expression of BECN1, BNIP3, and PPARGC1A was verified using tag-specific antibodies (Myc, FLAG, HA; white). Insets show higher-magnification views of BAG3-, SQSTM1-, or PINK1-positive puncta within myotubes. Scale bars = 10 μm. **b** Quantification of SQSTM1 immunofluorescence intensity in primary skeletal muscle cells after overexpression of BECN1 ($n = 15 - BAF$, $n = 13 + BAF$), BNIP3 ($n = 16 - BAF$, $n = 14 + BAF$), or PPARGC1A ($n = 16 - BAF$, $n = 11 + BAF$), compared with CTRL ($n = 10 - BAF$, $n = 11 + BAF$), showing increased autophagy, particularly after BECN1 overexpression. **c** Quantification of PINK1 immunofluorescence intensity after overexpression of BECN1 ($n = 10 - BAF$, $n = 10 + BAF$), BNIP3 ($n = 10 - BAF$, $n = 11 + BAF$), or PPARGC1A ($n = 11 - BAF$, $n = 11 + BAF$), compared with CTRL ($n = 11 - BAF$, $n = 11 + BAF$), indicating increased mitophagy, most pronounced after BNIP3 overexpression. **d** Quantification of BAG3^P209L-eGFP fluorescence intensity after overexpression of BECN1 ($n = 15 - BAF$, $n = 13 + BAF$), BNIP3 ($n = 16 - BAF$, $n = 14 + BAF$), or PPARGC1A ($n = 16 - BAF$, $n = 11 + BAF$), compared with CTRL ($n = 10 - BAF$, $n = 11 + BAF$), showing increased fluorescence, particularly after BECN1 overexpression with BAF. **e** Quantification of myotube area after overexpression of BECN1 ($n = 12 - BAF$, $n = 10 + BAF$), BNIP3 ($n = 11 - BAF$, $n = 10 + BAF$), or PPARGC1A ($n = 12 - BAF$, $n = 11 + BAF$), compared with CTRL ($n = 11 - BAF$, $n = 10 + BAF$), showing increased cell area, especially after BECN1 overexpression. **b**–**e** Data are presented as mean ± SD. n denotes independent myoblast differentiations derived from biologically independent mice. Statistical analysis was performed using a mixed-effects model (REML) followed by Šídák's multiple comparisons test. Source data are provided as a Source Data file.

muscle atrophy led to a 3-fold loss of strength in 4-week-old mice, as measured by ex vivo force measurements after electrical stimulation. Despite normalizing the stimulated force to muscle length and weight, both of which were significantly reduced in CAG-BAG3^P209L mice, force still remained significantly lower. From these data, it was evident that the lack of force development in skeletal muscles from CAG-BAG3^P209L mice was due to loss of sarcomeric structures. Another factor contributing to muscle weakness was impaired mitochondrial function, as mitochondrial energy supply and the ability to produce ATP were significantly decreased. This combination of structural damage and energy depletion explains the decreased force development in skeletal muscle from CAG-BAG3^P209L mice. This impairment correlated with the expression level of the BAG3^P209L-eGFP transgene and accounts for the phenotypic variability observed in this mouse line. The reason for the different expression levels of the transgene is unclear, as all mice analyzed were homozygous for the activated BAG3^P209L-eGFP expression.

To our surprise, the phenotype in skeletal muscle of CAG-BAG3^P209L mice and its underlying molecular mechanism were markedly different from those observed in the hearts of this mouse model[6]. Unlike cardiac muscle, skeletal muscle has a high regenerative capacity, and we demonstrated that the number of satellite cells was unchanged in CAG-BAG3^P209L-mice, implying that a regenerative response could still occur, as underscored by the increased number of skeletal muscle cells with centralized nuclei. Accordingly, in transcriptomic and proteomic analyses, enriched GO-terms included amino-acid synthesis, translation, and protein expression, which are clear signs of regeneration. Genes involved in ribosome biogenesis and assembly were enriched in skeletal muscle, as well as those involved in general protein translation and its initiation, and cytoplasmic translation. Proteomics analysis showed that the levels of structural proteins in the sarcomere, myofibril, Z-disc, and actin cytoskeleton were significantly reduced in the skeletal muscle of CAG-BAG3^P209L-mice. This suggests an inadequate induction of protein synthesis, which was unable to compensate for the loss of these structural proteins. Similar to the phenotype observed in cardiomyocytes of CAG-BAG3^P209L-mice[6], we found protein accumulation in the cytoplasm of skeletal myocytes and sarcomere disintegration. However, the total amount of BAG3 protein was 3 times lower in skeletal muscle compared to the heart, which could explain why the phenotype in the heart was more severe than in skeletal muscle. Increased autophagy was evident by vesicles observed in ultrastructural images and by higher expression of SQSTM1 and LC3BI and II. This indicated a blockage in autophagy and buildup of autophagic proteins, which explains the massive accumulation of proteins in skeletal myocytes. Additionally, expression of proteasomal genes were strongly enriched, as could be seen in RNA-Seq and proteomics analyses. Also, genes related to intrinsic apoptosis were enriched, along with genes involved in the response to oxidative

stress, reduced oxygen levels, and ER stress. This confirmed the increased apoptosis rate observed in EDL and soleus muscles (Supplementary Fig. 5c, d), further contributing to muscle atrophy. In summary, instead of a massive replacement fibrosis as seen in cardiac muscle, we detected signs of muscle regeneration in skeletal muscle and only mild fibrosis. Genes associated with mitophagy were highly enriched in transcriptomic and proteomic analyses of skeletal muscle, implicating accumulation of damaged mitochondria and a blockage of mitophagy.

Mitochondrial impairment was obvious in our biochemical oxygraphy analysis, and also in RNA-Seq data, as the GO-term "mitochondria depolarization" was enriched. Moreover, mitophagy factors such as PINK1 and BNIP3 were found to be enriched in the proteomics analysis of the insoluble fraction. This was further confirmed by electron microscopy, as mitochondria displayed a polymorphic shape and were found to accumulate in clusters, similar to reports in patients, suffering from MFM6[2] and in a mouse model with cardiomyocyte-specific BAG3^P209L overexpression[3]. In turn, impairment and blocked renewal of damaged mitochondria lead to lower ATP production and reduced force development in skeletal muscle.

To test whether disease progression can be halted, we used a gene therapy approach based on a shRNA that specifically silences the pathogenic hBAG3^P209L-eGFP fusion protein. Functionally, we found a significant improvement in muscle strength and an increase in muscle length and weight, proving that the gene therapy approach can prevent disease progression to a remarkable extent. This was underscored by the histological data, which demonstrated that the number of BAG3^P209L-eGFP-positive aggregates decreased in skeletal muscle cells transduced with AAV. Therefore, knockdown-based gene therapy works in principle for this disease and for myofibrillar myopathies in general, but an allele-specific siRNA must be identified before it can be used in patients. This also paves the way for more advanced therapy approaches, such as base editing or prime editing.

Since blockage of autophagy or mitophagy could be responsible for developing the phenotype, we tested this in skeletal myotubes derived from isolated primary skeletal muscle myoblasts from CAG-BAG3^P209L mice. This model system proved very well suited for these experiments, as BAG3^P209L-eGFP positive aggregates were observed. Importantly, only the induction of autophagy by BECN1, not the induction of mitophagy through overexpression of BNIP3, was able to decrease phenotypical features, such as SQSTM1 and BAG3^P209L-eGFP positive aggregates. We confirmed this in vivo by inducing autophagy through treating CAG-BAG3^P209L-mice with rapamycin, a well-known inhibitor of mTORC1. The treated mice exhibited significantly improved motor function, supporting the essential role of autophagy in developing the pathological phenotype.

Based on our experimental findings, we propose the following pathomechanisms underlying the skeletal muscle phenotype:

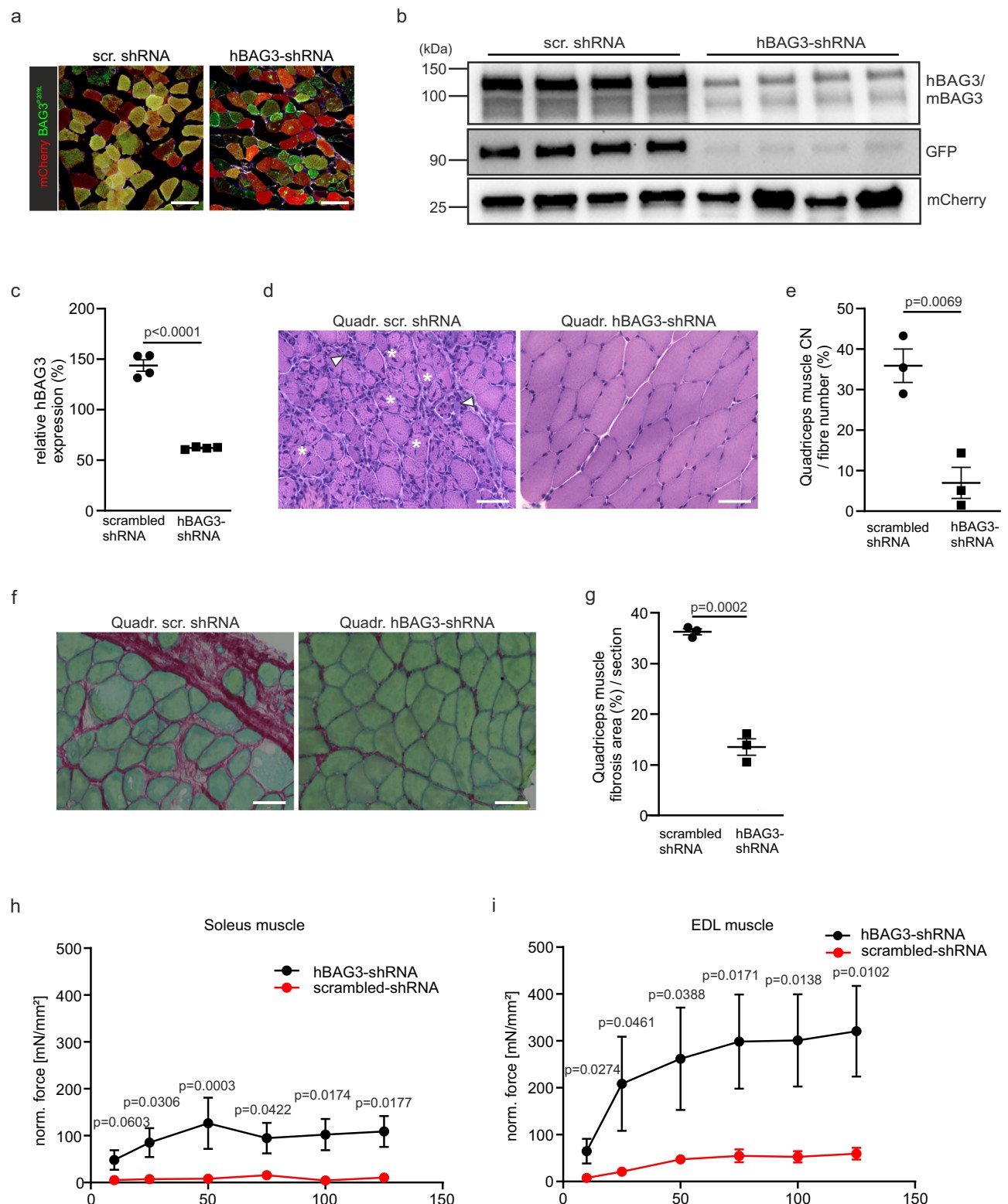

BAG3[P209L] has lower solubility than unmutated BAG3[WT] and, as reported earlier for the heart, will accumulate over time, forming cytoplasmic inclusions that also contain other proteins. This leads to three effects: (i) the loss of BAG3 function in the CASA complex as it gets withdrawn from the available pool, causing a blockage of autophagy, (ii) resulting in a reduced turnover of sarcomeric proteins mediated by BAG3, and (iii) impairment of mitophagy, which is secondary to the block in autophagy, causing an

accumulation of functionally impaired mitochondria. This will lead to the disintegration of sarcomeres, cell death, and reduced energy supply. Additionally, the BAG3-containing aggregates we observed could cause some of the structural changes by sterically hindering the contraction of the sarcomeres.

In summary, we demonstrate that even in skeletal muscle, which can regenerate, BAG3[P209L] causes a primary blockage in autophagy and a secondary blockage in mitophagy. This leads to sarcomere

**Fig. 8 | Improvement of the pathological phenotype after AAV-based gene therapy in vivo. a** Cross-sections from quadriceps muscle from AAV/rh10-treated CAG-BAG3$^{P209L}$ mice at P37: hBAG3$^{P209L}$-eGFP expression and aggregates were reduced after treatment with AAV/rh10 hBAG3, but not with scrambled shRNA. Scale bars: 50 μm. **b**, **c** Immunoblot analysis and quantification of protein expression showed a decrease in the amount of hBAG3 (upper BAG3 band) in CAG-BAG3$^{P209L}$ mice treated with AAV/rh10 hBAG3 shRNA compared to mice treated with AAV/rh10 scramble shRNA. Total protein staining was used to normalize expression. Mean ± SEM. $n = 4$ biologically independent quadriceps muscles per group. Statistical test = two-sided unpaired Student's $t$-test. scr. shRNA = scrambled RNA. **d** HE-staining of quadriceps muscle from 5 weeks old CAG-BAG3$^{P209L}$-mice which were treated with either AAV/rh10 hBAG3 shRNA or AAV/rh10 scramble shRNA at P15. Scale bar: 50 μm. **e** Quantification revealed a significant decrease in centralized nuclei from quadriceps muscle after treatment with AAV/rh10 hBAG3 shRNA in comparison to mice treated with AAV/rh10 scramble shRNA. Statistical test = two-sided unpaired Student's $t$-test, Mean ± SEM, $n = 3$ biologically independent

quadriceps muscles per group. **f** Analysis of quadriceps muscle fibrosis by Sirius red and Fast green staining after treatment with either AAV/rh10 BAG3 shRNA or AAV/rh10 scramble shRNA. **g** Quantification of the fibrotic area in quadriceps muscle from CAG-BAG3$^{P209L}$-mice 3 weeks after treatment with either scrambled or hBAG3 shRNA. Statistical test = two-sided unpaired Student's $t$-test, Mean ± SEM, $n = 3$ biologically independent quadriceps muscles per group. scr. shRNA = scrambled RNA. **a**–**g** Experiments were repeated three times with biologically independent replicates and yielded similar results. **h**, **i** Normalized force as determined by frequency-dependent force measurement (FFR) of soleus muscle (**h**) and EDL (**i**) from CAG-BAG3$^{P209L}$ mice after treatment with either AAV/rh10 BAG3 shRNA or AAV/rh10 scramble shRNA at P15. Sol.: $n = 3$ biologically independent hBAG3 shRNA mice; $n = 4$ biologically independent scr. shRNA mice; EDL: $n = 4$ biologically independent hBAG3 shRNA mice; $n = 6$ biologically independent scr. shRNA mice. Shown is mean ± SEM, statistical test = two-way repeated-measures ANOVA, followed by Šídák's multiple-comparisons test. Source data are provided as a Source Data file.

disintegration and mitochondrial dysfunction, causing muscle atrophy and impaired force development. Our successful shRNA-mediated knockdown illustrates that the mice serve as a useful model to test novel gene therapy approaches to heal this devastating disease.

## Methods

### Animal procedures
All applicable international, national, and/or institutional guidelines for the care and use of animals were followed. The reporting of animal experiments abides by the ARRIVE guidelines. All procedures performed in studies involving animals were following the ethical standards of the institution at which the studies were conducted and were approved by the responsible governmental authorities (Landesamt für Natur, Umwelt und Verbraucherschutz, NRW, Germany) (animal protocols #81-02.04.2021.A068 and 81-02.04.2019.A062). Mice are housed in a ventilated animal house (15 air changes per hour) in laboratory rooms under artificial lighting (12 h light/12 h dark), constant ambient temperature (22 °C), constant humidity (approximately 50%) with access to food and clean water ad libitum. Mice are fed standard laboratory animal feed (V1534-300 From ssniff Spezialdiäten GmbH, Germany) and maintained in approved laboratory cages 15 cm high and at least 400 cm$^2$ in size (5 individuals per breeding cage). Animals are not housed individually because, as preliminary studies have shown, this generates very severe stress, which may lead to their premature death. In order to provide mice with a pathological phenotype with appropriate conditions, the cage equipment is adapted to their current condition (appropriate bedding, food fed directly into the cage, areas providing shelter, appropriate attractants). In addition, periodic health evaluations by qualified persons is performed. Mice were killed by cervical dislocation.

Control mice (CTRL) were defined as siblings of homozygous PGK-Cre/CAG-flox-hBAG3$^{P209L}$-eGFP or PGK-Cre/CAG-flox-hBAG3$^{WT}$-eGFP mice with either WT, PGK-Cre, CAG-flox-hBAG3$^{P209L}$-eGFP or CAG-flox-hBAG3$^{WT}$-eGFP genotype. Mice from the homozygous (PGK-Cre/CAG-flox-hBAG3$^{WT}$-eGFP) strain did not display any obvious phenotype despite the strong hBAG3$^{WT}$-eGFP expression in their skeletal muscles. Generation of the CAG-flox-hBAG3$^{P209L}$-eGFP and the CAG-flox-hBAG3$^{WT}$-eGFP mouse lines has been described before[6].

The transgenic mice were bred in a mixed genetic background (129S6/SvEvTac x C57BL/6 N x CD-1). For genotyping of the Tg(CAG-flox-hBAG3$^{WT}$-eGFP) and Tg(CAG-flox-hBAG3$^{P209L}$-eGFP) transgenic mice genomic DNA of tail tips was isolated and PCR was performed using the primers CAG2-fw and BAG3-h-spez.tev, oIMR 9021 common rev and oIMR 9020 wt fw, and loxP-hygro-genotyp-BAG3 fw and BAG3 human-spez rev1 (see Supplemental Information Table 1 for primer information).

### Rapamycin/vehicle or saline injection
The experiments were conducted on 16 male 3 weeks old CAG-BAG3$^{P209L}$ mice with a strong phenotype, manifested by severe myofibrillar myopathy and cardiomyopathy and 8 male 3 weeks old non-transgenic littermates, constituting the control group (CTRL).

A dose of rapamycin of 8 mg/kg b.w., suspended in 0.9% NaCl and 2% ethanol at concentration of 0.5 mg/mL (547 μM), was used for experiments. During adaptation period, the mice were habituated to the administration procedure in order to minimize stress that could affect the obtained results. In the course of injection, the mouse was gently restrained (lifting the loose skin on the back of the neck) and a small-gauge needle (27 G) was inserted at a 45° angle in the lower right quadrant of the abdomen, avoiding the *cecum*. The vehicle and saline were administered in a similar manner.

### Sarcomere analysis multitool SarcAsM
To analyze sarcomere organization, we utilized SarcAsM, an AI-powered software designed for the automated, multiscale assessment of sarcomeres in microscopy images of myocytes immunostained for the Z-band protein ACTN2. For each myocyte, represented as a Z-stack of images, sarcomere and Z-band lengths were measured in each individual slice. These slice-by-slice measurements were then aggregated to calculate the mean and standard deviation for the entire cell, providing a quantitative measure of sarcomere and Z-band morphology. In a typical Z-stack, our analysis identified and measured an average of 540 distinct Z-bands (defined as structures >0.2 μm in length) and 400 sarcomere vectors, which represent the length and orientation of sarcomeres at different positions within the cell. To quantify the density of organized sarcomeric structures, we calculated the sarcomere area ratio. This was determined by creating maximum intensity projections of both the cell area mask and the sarcomere mask predicted by SarcAsM, and then computing the ratio of the sarcomere mask area to the cell mask area. Additionally, we assessed the cell area ratio, which represents the proportion of the total image area occupied by cells.

### Measurement of locomotor activity in actometers
The animals' locomotor activity was measured using actometers (Opto Varimex, Minor-Columbus, USA). The actometers consisted of four plexiglass walls measuring 43 × 43 × 20 cm. When the animal moved, a photocell recorded each interruption of the infrared beam, which was then counted by a digital counter. The animals' locomotor activity was recorded for 15 min and expressed as the number of horizontal movements (performed in the horizontal plane). Measurements were taken at a fixed time between 4.00 p.m. and 6.00 p.m. This test allowed us to determine the effect of rapamycin on improving the locomotor activity of CAG-BAG3$^{P209L}$ mice. The measurement was performed

three times: at point "0", after 7 and 14 days of rapamycin administration.

## Grip strength measurement

This test was conducted using a device that measures the grip strength of rodents. The mouse, held by the base of its tail, was first directed toward a metal cylinder, on which it rested its front paws, and then grabbed a rod used to measure grip strength with its hind paws. When the mouse grabbed the rod with its hind paws, it was pulled away from the grip until it lost its hind paw support. When the mouse released the grip, the maximum force value was recorded on the digital display of the control device. The measurement was performed three times: at point "0", after 7 and 14 days of rapamycin administration.

## The rota-rod test

Motor coordination was assessed using a rota-rod treadmill (Yamato Instruments Corporation, India). Before the proper measurement, the mice were subjected to a habituation procedure. At first, they were placed on a stationary rod for 120 s. Then, they were placed three times on a rod rotating at a speed of 3 rpm in a clockwise direction. The relevant measurement included the average of three trials of the time spent on a rod rotating at a speed of 30 rpm. The maximum duration of a single trial was 120 s. The measurement was performed three times: at point "0", after 7 and 14 days of rapamycin administration.

## Ex vivo skeletal muscle force measurement

Mice were killed by cervical dislocation. Dissection of hind limbs was performed in electrolyte solution permanently gassed with carbogen. Removing The skin was removed from the hind limbs and the distal common end tendon of the gastrocnemius and soleus muscles were detached from the calcaneus. By elevating the entire muscle bundle in a proximal direction, the soleus muscle was exposed with its tendon of origin. This tendon of origin was separated from the bone and the soleus muscle was separated from the gastrocnemius muscle in the region of the end tendon. Fixation of the hind legs with the dorsal side up, to remove the extensor digitorum longus (EDL) muscle. After removal of the connective tissue and ligaments in the area of the knee joint, the tendon of origin of the EDL was cut directly at the bone. The soleus muscle and the EDL muscle were clamped at their tendons into the Myostation of the Myotronic device to determine the force of the skeletal muscle. The muscles were set to their optimal muscle length and were in a permanently carbogen-gassed electrolyte solution during the measurement. To analyze the relationship between stimulation frequency and muscle force development in mN, stimulation was first performed every 3 min at 10 Hz (soleus muscle) or every 5 min at 10 Hz (EDL), followed by a pause of 3 (soleus muscle) or 5 min (EDL muscle) and then a second stimulation at 25 Hz, followed by 50 Hz, 75 Hz, 100 Hz and the final frequency of 125 Hz. After completion of the measurements, the length of the purely muscular part of the muscle was determined and the non-muscle parts were removed so that the muscle could be weighed.

## Histology, immunofluorescence staining, and microscopy

Skeletal muscles were dissected and fixed in 4% paraformaldehyde (PFA) at 4 °C for 4 h. The fixed tissue was equilibrated in 20% sucrose solution and then properly oriented and embedded in Tissue Tek® O.C.T Compound. 10 μm thick cryosections were cut on a cryotome CM 3050S (Leica). Sirius red and Fast green staining was performed using standard histology protocols. For staining of fixated cells and skeletal muscle sections the following antibodies were used (for 2 h at RT in 0.2% Triton X in PBS, 5% donkey serum or overnight at 4 °C in 0.2% Triton X in PBS, 5% donkey serum): ACTN2 (1:400, Sigma-Aldrich), BAG3 (1:250, Proteintech, recognizing both human and mouse BAG3), desmin (1:100, DAKO), FLNC (1:1000, Biogene), SYNPO2 (1:100, Invitrogen), cCasp3 (1:50, Cell Signaling Technology), CD45

(1:400, Merck Millipore), BF-F3 (1:400, Developmental Studies Hybridoma Bank (Iowa, USA)), BA-F8 (1:400, Developmental Studies Hybridoma Bank (Iowa, USA)), SC-71 1:400 (Developmental Studies Hybridoma Bank (Iowa, USA)), LC3B (1:300, Cell Signaling), SQSTM1 (1:300, Progen), Pax7 (1:50, Developmental Studies Hybridoma Bank (Iowa, USA)), Ki67 (1:200, Dako), WGA (1:1000, Vector Laboratories). After washing, secondary antibodies conjugated to Alexa Fluor 488, Cy3 or Alexa Fluor 647, respectively (1:400, Jackson ImmunoResearch), diluted in 1 μg ml$^{-1}$ Hoechst (Thermo Scientific, 33342) in PBS were applied for 1 h at room temperature. Sections were mounted with coverslips and imaged using an inverted fluorescence microscope (Axiovert 200, Carl Zeiss) with filters for DAPI, GFP, Cy3, and Cy5, and ×25, ×40, and ×60 Plan Apochromat water objectives, an ebx 75 light source and an AxioCam MRm digital camera. The editing of the pictures was performed with the Axiovision Rel. 4.8 Software (Zeiss).

## Protein isolation

Tissues from PGK-Cre × CAG-BAG3 (P209L)-eGFP-flox mice and controls were prepared with 8M-urea buffer to extract the proteins within. Due to the instability of urea, the urea buffer always had to be freshly prepared. 50 mg of tissue was mixed with 500 μl of urea buffer and homogenized with a pestle. Protein concentration was determined using the PierceTM BCA Protein Assay Kit (Thermo Scientific, (Waltham, USA)). Quantification of protein concentration of each sample was determined using a microplate reader (Tecan Spark Reader, Cat. No. 30086376) with a measurement below 562 nm.

## Protein isolation from quadriceps muscle for enzyme-linked immunosorbent assay (ELISA)

The mice were administered a lethal intraperitoneal dose of pentobarbital anesthesia at a dose of 120 mg/kg b.w., and then the quadriceps muscles were collected. To minimize discomfort to the animal during this procedure, the mice were additionally anesthetized with isoflurane (2.5%, flow rate 0.5 L/min) immediately before the injection. Quadriceps muscles were taken from both hind limbs, and the muscles were cleaned of fat and connective tissue. The collected tissues were frozen using liquid nitrogen and then stored at −80 °C until further analysis. Next, the quadriceps muscles were divided into pieces weighing 100 mg each and placed in microcentrifuge tubes together with an equal mass of zirconium silicate beads (0.5 mm; Next Advance Inc., ZROB05). Then, T-PER Tissue Protein Extraction Reagent was added in a volume ratio of 2:1 to the volume of the tissue (max. 0.6 ml of the homogenization buffer). The tube was sealed and placed in a Bullet BlenderTM, the speed was set to 8, and the time was set to 4 min.

## Western blot analysis and enzyme-linked immunosorbent assay (ELISA)

Specific proteins were separated according to their size in a denaturing SDS-polyacrylamide gel. 30 μg of the samples were mixed with the appropriate amount of 4× Laemmli Sample Buffer (Bio-Rad, cat#161-0747) and then denatured at 99 °C for 10 min. The 4–20% Mini-PROTEAN® TGX Stain-Free™ Protein gel (BioRad) was placed in the gel electrophoresis chamber and filled with 10× Tris/Tricine/SDS Running Buffer (BioRad, cat#1610732). The first well of the polyacrylamide gel was filled with molecular weight marker (BioRad, cat#1610376), and the remaining wells were successively filled with the prepared samples.

After size-dependent separation of the proteins using the SDS-polyacrylamide gel, they were blotted to a polyvinylidene fluoride (PVDF) membrane (BioRad, cat#1704274) at 25 volts for 10 min using the Trans-Blot Turbo Transfer System (BioRad). Immunostaining was preceded by blocking the nonspecific binding sites by incubating the membrane in 5% skim milk powder in TBS-T for 1 h. The primary antibody was also prepared in 5% skim milk powder in TBS-T (Bag3 1:5000, CRYAB 1:5000, HSPB6 3:5000, HSPB7 1:1000, HSPB8 1:1000, SQSTM11:1000, LC3B 1:1000, SYNPO2, ß-actin 1:2000, GFP 1:1000,

mCherry 1:1000) or in 5% BSA in TBS-T (PINK1 1:1000). Incubation of the membrane with the first antibody was performed overnight at 4 °C. The membrane was then washed three times with 6 ml of TBS-T each time, with TBS-T removed directly the first time, followed by two five-minute incubations. The secondary antibodies (HRP, horseradish peroxidase or fluorescence-coupled) were diluted in 5% skim milk powder/BSA in TBS-T and incubated with the membrane for 1 h at room temperature. The membrane was then washed five times with TBS-T for 5 min each time. The PierceTM Chemiluminescent Reagent (ECL Western Blotting Substrate) from ThermoScientific was used to detect HRP-coupled secondary antibodies. The membrane was treated for 1 min with a 1:1 mixture of the peroxide solution and the luminol enhancer solution of the ECL system. The ChemiDocTM MP imaging system from BioRad was used for detection. Analysis of the acquired images was performed using BioRad's Image Lab software, normalizing the relative protein expression of each sample to β-actin or total loaded protein.

The level of LC3 II protein was determined in homogenates from quadriceps muscles and hearts using the ELISA method (cat. number: MBS3806182; MyBioSource, Inc.; San Diego, USA).

### Isolation and transfection of primary myoblasts from mice

Primary myoblasts were isolated from hindlimb muscles (tibialis anterior, gastrocnemius, soleus, quadriceps, and extensor digitorum longus) of 6-week-old mice following euthanasia by cervical dislocation. Muscles were aseptically excised, cleared of fat and connective tissue, repeatedly washed, and finely minced under sterile conditions. Tissue was enzymatically digested with collagenase II at 37 °C for 50 min, followed by mechanical dissociation, sequential filtration (70 μm, then 30 μm mesh), and centrifugation. Cells were resuspended in myoblast growth medium supplemented with bFGF and seeded onto 10% Matrigel-coated 100-mm culture dishes, maintained at 37 °C and 5% CO$_2$. Fibroblasts were depleted by repeated pre-plating on uncoated dishes until a myoblast purity of >95% was achieved. Cells were maintained below 75% confluency, with medium changes every 2 days. For differentiation, myoblasts were cultured to 75–80% confluency, after which growth medium was replaced with myotube differentiation medium, refreshed every 48 h. Myoblast alignment and fusion were observed within 24–36 h, with initial myotube formation after ~48 h and further maturation over 72–96 h. After 6 days, myotubes exhibited advanced maturation with clearly visible cross-striations. Mature myotubes were transiently transfected with plasmids carrying expression cassettes for *Becn1*, *Bnip3*, or *Ppargc1a* using Lipofectamine 3000 (Thermo Fisher). On the day of transfection, the culture medium was replaced with fresh differentiation medium without penicillin/streptomycin (500 μl per well). For each well of a 24-well plate, 1 μg plasmid DNA was diluted in 25 μl Opti-MEM reduced-serum medium (Thermo Fisher Scientific) and mixed with 1 μl P3000 (Thermo Fisher Scientific) reagent. In parallel, 1.5 μl Lipofectamine 3000 (Thermo Fisher Scientific) was diluted in 25 μl Opti-MEM and incubated for 5 min at room temperature. DNA and Lipofectamine solutions were combined and incubated for 10–15 min at room temperature to allow complex formation. The transfection mixture was added dropwise to the cells, which were then incubated at 37 °C and 5 % CO$_2$. After 17 h, transfection was terminated by replacing the medium with fresh differentiation medium. At 48 h post-transfection, cells were either fixed with 4% paraformaldehyde or treated with 300 nM Bafilomycin A1 for 5 h, followed by a 1 h incubation in differentiation medium prior to fixation. BECLIN-HA WT was a gift from Kunliang Guan (Addgene plasmid # 46993; http://n2t.net/addgene:46993; RRID:Addgene_46993). AAV-CMV-Flag-PGC1a-6His was a gift from Connie Cepko (Addgene plasmid # 67637; http://n2t.net/addgene:67637; RRID:Addgene_67637). Myc-Bnip3FL was a gift from Joseph Gordon (Addgene plasmid # 100796; http://n2t.net/addgene:100796; RRID:Addgene_100796).

### Allele-specific *BAG3* expression-analysis

RNA was isolated from quadriceps muscles of 4 weeks old CAG-BAG3$^{WT}$, BAG3$^{P209L}$ or non-transgenic sibling control mice using the RNeasy Plus Micro Kit (#74034, Qiagen) according to manufacturer's instructions. The RNA-integrity was measured with the Bioanalyzer 2100 and RNA 6000 Nano Kit (#5067-1511, Agilent). Only samples with RNA-integrity number (RIN) above 6.4 were used for further experiments. 500 ng of RNA of each sample were transcribed to cDNA in 20 μl mixtures using the SuperScript VILO Master Mix (#11755500, Thermo Fisher) according to manufacturer's instructions. For allele-specific *BAG3* expression analysis an SNP Genotyping TaqMAN qPCR-Assay (Assay-ID: C_160294934_10, SNP-ID: rs121918312, Thermo Fisher) from was used which includes two highly allele-specific fluorescent-labeled probes for *BAG3* WT and *BAG3* P209L (*BAG3* c.626 C/T). In separate mixtures TaqMAN probes for murine *Bag3* (Assay-ID: Mm00443474_m1Bag3, Thermo Fisher) and murine *Gapdh* (Assay-ID: Mm99999915_g1, Thermo Fisher) were used as references. The qPCR was prepared in 20 μl format and technical duplicates using 25 ng of cDNA, 10 μl TaqMAN Master-Mix for gene expression (#4369016, Thermo Fisher) from, either 1 μl of SNP Genotyping TaqMAN probes or 1 μl of each of the TaqMAN probes against murine *Bag3* and murine *Gapdh*. H$_2$O was added until 20 μl for each mixture. The qPCR was performed on the CFX Opus 96 Real-Time PCR System from Bio-Rad using following run-parameters: Initial denaturation 95 °C 10 min; 40 cycles: Denaturation 95 °C 15 s, annealing/polymerization 60 °C 1 min. For further analysis the thresholds of the two fluorophore-channels had to be set to the same value by hand to allow allele-specific comparisons. Relative expression was calculated using RE = 2$^{-(\Delta Cq)}$ while $\Delta Cq = Cq_{BAG3\text{-of-interest}} - Cq_{mGAPDH}$.

### Fractionation of mouse muscle tissue

Mouse quadriceps muscles were used for the proteomics analysis ($n = 4$ for each genotype). Each muscle sample was cut into 100 thin sections of 5 μm thickness in a cryotome at −20 °C. The thin sections were taken in 2 mL tubes containing 1.4 mm and 2.8 mm zirconium oxide beads and weighted. 1× Triton X–100 buffer (Cell Signaling Technology) containing HALT™ protease and phosphatase inhibitors (Thermo Scientific) at a ratio of 1:100 (v/v) was added at an amount of 30 μL per milligram of the sample. The samples were homogenized using a Percellys 24 tissue homogenizer (Bertin instruments). The homogenization was done in 3–5 cycles, each cycle consisting of homogenization at 5800 rpm for 15 s followed by resting on ice for 120 s. About 50 μL of the lysed sample was collected in a micro-centrifuge tube, and a 100 mM Tris−HCL buffer (pH 7.9) containing 4 M urea, and HALT protease and phosphatase inhibitors at 1:100 (v/v) ratio was added. The remaining sample was centrifuged at 20,800×$g$ for 15 min, and Triton X-100 soluble supernatant was transferred to a new microcentrifuge tube. The remaining pellet was washed with Triton buffer, centrifuged, and transferred to the rest of the supernatant and stored at −80 °C. The remaining Triton X−100 resistant pellets were solubilized using the above-mentioned urea buffer at a volume equal to 70% of the initial Triton X−100 buffer. The samples were stored at −80 °C until further processing.

### Proteomics sample preparation

About 20 μg of the proteome sample from the total lysate, the pellet and the supernatant fraction ($n = 4$ biological replicates for each condition and each of the two genotypes) were carbamidomethylated by incubating with 10 mM Dithiothretol (DTT) at 37 °C for 30 min followed by incubating with 50 mM Chloroacetamide for 30 min in the dark at room temperature. After that the reaction was quenched by adding 50 mM DTT. The proteome was purified using SP3 beads as described earlier[16], dissolved in 100 mM HEPES-NaOH buffer of pH 7.5 and 2.5 mM CaCl$_2$ and digested with trypsin at a ratio of 1 μg trypsin to 100 μg proteome over night at 37 °C, shaking. Digested peptides were

desalted using self-packed StageTips following the protocol as described by Rappsilber et al.[17]. The eluted peptides from the Stage-Tips were dissolved in HPLC-water with 0.1 % formic acid and analyzed using LC-MS/MS setup.

## Mass spectrometry data acquisition

The peptides were separated using an Ultimate 3000 RSLC nano chromatography system (Thermo Fisher) operated in two-column setup. First, the peptides were loaded on to a short 2 cm μPAC reverse phase trap column (Thermo Fisher) using a loading pump buffer of 2% acetonitrile and 0.05% TFA. The peptides from the trap column were eluted and separated in a 50 cm μPAC reverse phase analytical column (Thermo Fisher) with an acetonitrile gradient of 2 to 30% eluent B (Eluent A−HPLC water, 0.1% FA, Eluent B−acetonitrile, 0.1% FA) over a period of 90 min. The whole LC method for each sample was about 120 min including the washing and the equilibration phases.

Peptides eluted from the column were directly introduced into an Impact II Qq-ToF-MS (Bruker) using a captive spray ion source as described[18]. The instrument was operated in data independent acquisition (DIA) mode, with acquisition of MS1 spectra from 400 m/z to 2200 m/z followed by MS2 spectra of ions in 32 mass windows of 25 m/z from 400 m/z to 1200 m/z. Each mass window had an overlap of 0.5 m/z on either site to compensate for the inefficiency of the quadrupole. The MS was operated at 15 Hz resulting a cycle time of 2.2 s.

## Proteomics data analysis

The acquired data was converted into mzML format using proteowizard[19] with the recommended settings for DIA-NN analysis. The mouse reference proteome was downloaded from Uniprot[20] (release 2022_05). A FASTA database listing contaminants commonly observed for mouse tissue was downloaded from the github (https://github.com/HaoGroup-ProtContLib) and the BAG3$^{P209L}$-EGFP protein sequence was also added to the search. DIA-NN[21] version 1.8 was used for peptide identification with the following settings: Methionine excision and carbamidomethyl were set as fixed modifications, enzyme was set to strictly trypsin with up to 1 missed cleavage, and variable modification was set to 1 and methionine oxidation was set a variable modification. The MS1 and MS2 mass error were set to 20 ppm with a scan window of 4. Deep learning-based spectra and RT prediction was switched on. The analysis mode was set to double pass mode and quantities matrices were switched on. All other settings were default. Data obtained from DIA-NN were further analyzed using Perseus[22] version 1.6.15.0. The protein groups matrix was uploaded to Perseus and the values were log2 transformed. After removing contaminants and non-unique genes, the matrix was filtered for valid values. Proteins found exclusively in samples of one genotype (quantified in at least 3 of the 4 replicates) were analyzed separately. For proteins found in both control and the mutant mice, significantly changing proteins were identified with an FDR controlled. GO and KEGG term enrichment analyses were done using cytoscape[23] and string DB[24]. In cytoscape ClueGo was used with standard settings and only significant pathways with a Bonferroni stepdown corrected $p$-value ≤ 0.05. Box plots were created using Raincloud Shiny app[25].

## Electron microscopy

A standard EM protocol was used in this study, briefly, stretched quadriceps femoris muscle samples from WT and CAG-BAG3$^{P209L}$-mice ($n$ = 3 per genotype) were fixed in 4% paraformaldehyde, 15% saturated picric acid and 0.5% glutaraldehyde in 0.1 M phosphate buffer pH 7.4 overnight at 4 °C. After rinsing in PBS, sections were treated with 0.5% OsO$_4$, washed and counterstained with uranyl acetate, dehydrated via ethanol series and embedded in Durcupan resin (Fluka, Switzerland). Ultrathin sections were prepared (Ultracut S; Leica, Germany) and adsorbed onto glow-discharged Formvar-carbon-coated copper grids. Ultrastructural analysis was performed in a Zeiss LEO 910 electron microscope and images were taken with a TRS sharpeye CCD camera (TRS Systems, Moorenweis, Germany).

## Isolation of skeletal muscle mitochondria

Mouse skeletal muscle mitochondria were prepared by rapidly removing the hindlimb muscles and transferring them into ice-cold isolation medium (180 mM KCl, 10 mM EDTA-Na2, pH 7.4). Muscles were minced with scissors, trimmed clean of visible fat and connective tissue, and placed in 30 ml of isolation medium supplemented with trypsin (1 mg per 1 mg of tissue). After 30 min, the tissue was homogenized using a motor-driven Teflon-glass Potter homogenizer. The homogenate was centrifuged at 800×$g$ for 6 min. The supernatant was decanted and centrifuged at 9000×$g$ for 10 min at 4 °C. The final mitochondrial pellet (M-fraction) was resuspended in medium containing 0.2 M sucrose and 20 mM HEPES, pH 7.2

## Determination of substrate oxidation rates

Mitochondrial oxygen consumption was measured at 25 °C using an Oroboros oxygraph (Anton Paar, Austria) in a medium containing 10 mM KH$_2$PO$_4$, 60 mM KCl, 60 mM Tris, 110 mM mannitol, 5 mM MgCl$_2$, 0.5 mM EDTA, pH 7.4 and 10 mM glutamate (glu) + 5 mM malate (mal) or 1 mM octanoylcarnitine (OC) + 5 mM malate (mal) as respiratory substrates. The respective additions were 10 mM creatine (dry powder), 1 μM carboxyatractyloside (CAT) or 2 μM of the uncoupler TTFB. The concentration of mitochondria was 0.3 mg protein/ml.

## Mitochondrial enzymatic measurements

We measured activity of the mitochondrial enzymes NADH-CoQ oxidoreductase, cytochrom c oxidase (COX), as well as citrate synthase using standard methods. All measurements were performed at 30 °C with a dual wavelength spectrophotometer (Aminco DW-2000, SLM Instruments, Rochester, NY, USA). Shortly, the activity of rotenone-sensitive NADH-CoQ1 oxidoreductase was measured in a reaction medium containing 50 mM KCl, 1 mM EDTA, 10 mM TRIS-HCl (pH 7.4), 1 mM KCN, 100 μM CoQ$_1$, and 150 μM NADH. The assay was initiated by addition of the sample and the velocity of NADH oxidation was monitored. After 2 min of registration, 20 μM rotenone was added to determine the rotenone-insensitive NADH oxidation rate. The activity of mitochondrial NADH-CoQ$_1$ oxidoreductase is the difference between the total NADH oxidation rate and the rotenone-insensitive NADH oxidation rate. The cytochrome c oxidase activity was measured in 0.1 M potassium phosphate buffer (pH 7.4), 0.02% laurylmaltoside and 200 μM reduced cytochrome c. The reaction was started with sample addition and the oxidation of ferrocytochrome c was monitoring at the wavelength pair 510/535 nm ($\varepsilon_{red-ox}$ = 5.9 mM$^{-1}$ cm$^{-1}$).

## Tissue RNA-seq analysis

For RNA-Seq experiments, DNA-free total RNA was isolated from quadriceps muscle from 4 CAG-BAG3$^{P209L}$ and 5 control mice (5 weeks old) using the RNeasy Kit (Qiagen) including on-column DNAse digestion. RNA quality was analyzed by an Agilent Bioanalyzer (Agilent). For library preparation the Trio RNA-Seq Library Preparation kit for mouse (TECAN) was used, starting with 50 ng of total RNA. Six PCR cycles were used for library amplification and libraries with an average fragment size of 322 bp were sequenced on a NextSeq 500 in paired-end mode (75 bp, Illumina). For bioinformatics analysis, we used the Galaxy platform (Freiburg Galaxy Project[26]). RNA sequencing reads were mapped using RNA STAR[27] followed by counting reads per gene by using featureCounts[28]. Differentially expressed genes were identified by DESeq2[29]. For data visualization, normalization and cluster analysis heatmap2 and Volcano plot (Freiburg Galaxy Project[28]) was used. Gene ontology analysis was performed by ClueGO by using the KEGG pathway and GO-term databases with a significance interval for pathways of $p$ < 0.05. $p$-Values were corrected for multiple-testing by the Holm-Bonferroni method.

## Injection of AAV/rh10

AAV/rh10-mCherry-U6-h-BAG3-shRNA (RNAi) and AAV/rh10-mCherry-U6-scrmb-shRNA (control) were used for gene therapy (Vector Bio-Labs). P15 CAG-BAG3$^{P209L}$ mice were injected into the left jugular vein with a total volume of 100 μl PBS containing $1 \times 10^{11}$ virus particles of either RNAi or control AAV/rh10. Mice were euthanized 3 weeks later (P37) and analyzed.

## Statistical analysis

Data are depicted as mean ± SEM. Normal distribution of data was determined by D'Agostino & Pearson omnibus normality test, while the Levene test was used to assess the homogeneity of variance. Depending on these results, analysis of variance (ANOVA) followed by Tukey's post hoc test was performed for normally distributed data. If the assumptions of normality and homogeneity of variance were not met, the nonparametric Kruskal-Wallis test was applied, followed by Dunnett's test for multiple comparisons. Unpaired $t$-test (two-sided) was used to test for significance for two independent groups. All statistical analyses were conducted using GraphPad Prism V9.0.1 software. $P$-values < 0.05 were considered statistically significant. Statistically significant differences are indicated by exact $P$-values. All measurements were taken from distinct samples.

## Reporting summary

Further information on research design is available in the Nature Portfolio Reporting Summary linked to this article.

## Data availability

The mass spectrometry proteomics raw data generated in this study have been deposited to the ProteomeXchange Consortium via the PRIDE[30] partner repository under accession code PXD047942. The tissue RNA-seq raw data used in this study are available in the SRA database under accession code PRJNA1082302. The mass spectrometry proteomics data and the tissue RNA-seq data generated in this study are provided in the Supplementary Information/Source Data file. Source data are provided with this paper.

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

## Acknowledgements

We thank the Freiburg Galaxy Team, Björn Grüning, Anika Erxleben, and Rolf Backofen, Bioinformatics, University of Freiburg, Germany, funded by the Deutsche Forschungsgemeinschaft (SFB 992 and SFB 1425,

project S3) and German Federal Ministry of Education and Research (BMBF grant 031 A538A RBC [de.NBI]). We thank the European Molecular Biology Laboratory GeneCore (Heidelberg, Germany) for providing sequencing services. This work was supported by the German Research Foundation (FOR2743 to P.F.H. and M.H. #388932620). B.K.F. is a member of CRC1425, funded by the German Research Foundation. The Galaxy server that was used for some calculations is in part funded by the Collaborative Research Center 992 Medical Epigenetics (DFG grant SFB 992/1 2012) and the German Federal Ministry of Education and Research BMBF grant 031 A538A de.NBI-RBC.

## Author contributions

K.F., K.G.R., K.K., J.R., and C.K. performed molecular biology, cell culture, imaging, and immunohistochemistry experiments and data analysis. W.A.L. and A.U. designed, performed, and analyzed (immuno)electron microscopy experiments. M.K., M.M., H.B., and P.F.H. designed and performed mass spectrometry and proteomics data analysis. A.P.K., C.K., and W.S.K. designed and performed isolation of skeletal muscle mitochondria, determination of substrate oxidation rates and measurements of mitochondrial enzymes. K.F. and K.G.R. performed ex vivo Skeletal muscle force measurement. M.P. and G.W. designed, performed and analyzed the rapamycin treatment experiment. D.H. performed analysis with the SarcAsM algorithm. M.W. performed qPCR experiments and data analysis. M.H. performed tissue RNA-Seq analysis. M.H. conceived the study and wrote the manuscript together with B.K.F. All authors edited and approved the submitted paper.

## Funding

## Competing interests

The authors declare no competing interests.
