## [Transparent Peer Review file · Nature Communications]

Blockage of autophagy causes severe skeletal muscle disruption in a mouse model for myofibrillar myopathy 6

Corresponding Author: Dr Michael Hesse

Version 0:

Reviewer comments:

Reviewer #2

(Remarks to the Author)

Strengths

The paper described an exhaustive characterization of an overexpression model of Myofibrillar myopathy-6 (MFM6). This is the first publication of such model. Producing a mice model of MFM6 has been extremely difficult since the KO are lethal and double KI for the P209L mutation do not show a phenotype and KI/KO only a very mild phenotype.

The paper makes all efforts to try to characterize clinically, histologically, biochemically and at the RNA expression level the impact of the overexpression of a human hBAG3P209L. The authors are not clear early on that these mice are homozygous for the insertion.

The sum of the work is impressive, and the results are well presented, and the paper is well written.

Weaknesses

There should have been a better description of previous efforts to develop a mice model of MFM6.

The rationale for generating a partially humanized, since the mice BAG3 are not inactivated, overexpression is not detailed in the introduction. This is important, since the major reservation for this model is that all the findings may mostly be caused by the overexpression of the hBAG3P209L. This is not well discussed in the paper and should be stated and discussed as one of the major limitations of the model.

The authors do not underline that very high levels of expression hBAG3P209L with an eGFP tag further suggests that it may have a larger effect on aggregating of the targets of Bag3 and wildtype Bag3.

More discussion on the relative tissue expression of the wildtype and hBAG3P209L should have been included to insist on the difference between heart and skeletal muscle.

A model with at the most a hBAG3P209L expression level comparable to the wildtype would have been preferable.

Questions to be addressed

Page 13: How significant was the "...increase of BAG3, α B-crystallin (Fig. 5c, d) and the small heat shock proteins HSPB7 and 8 (Fig. 5e, f), of which HSPB8 binds BAG3" should be stated in this sentence.?

Page 17: They provide strong evidence that overexpression of hBAG3P209L leads to the: "... 215 of proteins accumulating in the pellet fraction ($\log_2(\text{BAG3P209L/control}) > 0.25$) showed reduced abundance in the soluble fraction ($\log_2(\text{BAG3P209L/control}) < -0.25$), indicating increased association to the cytoskeleton or aggregation (Fig. 6d). More data on relative overexpression and phenotype should be included.

Page 21: Over expression is likely the key driver to phenotype, which is not what is observed in the human pathology, as its level influence severity as stated: "This impairment correlated with the expression level of the BAG3P209L-eGFP transgene and explained the phenotypic variability we observed in this mouse line. The reason for the different expression levels of the transgene are unclear as all mice analyzed were homozygous for the activated BAG3P209L-eGFP expression."

Page 22: This difference in expression between heart and skeletal muscle are important. Could we compare levels of expression in the mice?

Minor points

A supplemental figure with the alignment of the hBAG3P209L compared to the mice Bags3 would be appreciated.

Page 11 legend of figure 4: 20 most differentially expressed genes are shown. Not genes but pathways.

Page 12: adherents rather than "adherens"

Page 16: Figure d, the lines to dots for hBAG3P209L and mice Bag3 are not clear.

Reviewer #3

(Remarks to the Author)

The authors present a mouse model for BAG3P209L myofibrillar myopathy that shows molecular and functional features of severe muscle weakness as described in patients with MFM6. They demonstrate that accumulation of BAG3-containing aggregates and loss of BAG3 function leads to sarcomere disintegration, mitochondrial dysfunction and subsequently muscle weakness and this can be rescued by knockdown of hBAG3P209L.

While potentially interesting, the study is essentially uncontrolled because they did not generate and characterize a mouse line that overexpresses WT BAG3 to the same level. Overexpression of many WT proteins leads to phenotypes so there is no way here of knowing if the observed phenotypes were due to the P209L mutation or gross overexpression of BAG3.

Reviewer #4

(Remarks to the Author)

In the manuscript the authors studied the role of BAG3 mutation in muscle physiopathology. They generated a humanized transgenic mouse line that ubiquitously expressed the mutant BAG3P209L protein. These mice have been previously characterized from a cardiac point of view and indeed they showed a cardiomyopathy and died within 5-8 weeks. When the authors checked the skeletal muscle function and structure, they found the typical features of Myofibrillar myopathies (MFM) such as myofibrillar disassembly, protein aggregates and weakness. Transcriptomics and proteomics analyses confirmed an alteration in proteostasis with impairment of autophagy/mitophagy and accumulation of dysfunctional mitochondria. Almost all these features were reverted by a genetic approach to knockdown BAG3 expression in skeletal muscles. The paper is primarily descriptive of the muscle phenotype when BAG3P209L protein is expressed. The phenotype is very interesting but the insights that explain the muscle features with BAG3P209L are superficially explored. The authors should consider the following points

Point1. Figure 1f-g. The GFP pattern is not homogeneous among the different myofibers, especially in the soleus. Is there any fiber type specificity? From the pattern it looks like that it is higher expressed in the fast fibers. Authors should check the GFP expression in the different fiber types

Point2. Authors should perform morphometric analyses in different muscles such as Tibialis Anterior, Gastrocnemius and Diaphragm.

Point3. Fig2a. Check also Diaphragm, which is the most affected muscle in murine models of Muscle Dystrophy. Check also fat infiltration with Sirius red staining

Point4 Fig3c-d are of poor quality. Show high resolution images

Point5. Fig3e-g. EM pictures suggest accumulation of vesicles/lysosomes. Show high magnification pictures to underline the features. Authors should immunostain for Lamp1, p62 and LC3 and quantify the puncta.

Point6. Autophagy and mitophagy flux must be monitored to support the authors claims.

Point7. In vivo Bag3 knockdown results in phenotypic amelioration. Authors should also monitor whether mitochondrial function and autophagy was restored.

Point8. Author should put an effort in identifying which mechanisms downstream BAG3 dysfunction is the main trigger of the phenotype. Is autophagy reactivation sufficient to restore muscle function in BAG3P209L mutant? Or is mitochondrial improvement via mitophagy? Or just mitochondrial function that when preserved also improves muscle function. These experiments could be easily done (e.g. Beclin1 expression or rapamycin treatment for autophagy activation, BNIP3 expression for mitophagy, PGC1a expression for mitochondrial biogenesis and function) and would confirm/identify insights that connect BAG3 mutation to MFM,

Version 1:

Reviewer comments:

Reviewer #2

(Remarks to the Author)

The authors have made very significant modifications to the manuscript, including adding data on a new mice line that

further supports that the original line is a good model for myofibrillar myopathy 6 and not due to overexpression of the tagged transgene. They have made very constructive modifications in response to the different reviewers comments that have improved clarity. This is an important contribution to the field.

Reviewer #3

(Remarks to the Author)

Whilst the authors have made a valiant attempt to address my concern that the phenotypes associated with transgenic overexpression of mutant BAG3P209L might simply be due to overexpression and not the mutation, their control transgenic line only expresses WT BAG3 protein to 39% of the level achieved with BAG3P209L so unfortunately I do not think they can rule out simple overexpression as a source of any or all phenotypes.

Reviewer #4

(Remarks to the Author)

The authors addressed my requests. The paper has been improved

Point-by-point response to Reviewer Comments

Reviewer #2 (Remarks to the Author):

Strengths

The paper described an exhaustive characterization of an overexpression model of Myofibrillar myopathy-6 (MFM6). This is the first publication of such model. Producing a mice model of MFM6 has been extremely difficult since the KO are lethal and double KI for the P209L mutation do not show a phenotype and KI/KO only a very mild phenotype.

The paper makes all efforts to try to characterize clinically, histologically, biochemically and at the RNA expression level the impact of the overexpression of a human hBAG3P209L. The authors are not clear early on that these mice are homozygous for the insertion.

The sum of the work is impressive, and the results are well presented, and the paper is well written.

Answer: We thank the reviewer for appreciating our work and his/her thoughtful and constructive comments and suggestions.

Weaknesses

There should have been a better description of previous efforts to develop a mice model of MFM6. The rationale for generating a partially humanized, since the mice BAG3 are not inactivated, overexpression is not detailed in the introduction. This is important, since the major reservation for this model is that all the findings may mostly be caused by the overexpression of the hBAG3P209L. This is not well discussed in the paper and should be stated and discussed as one of the major limitations of the model.

Answer: As suggested by the reviewer, we have included a description of prior mouse models in the revised version of the Introduction. Furthermore, we provide a more detailed explanation of the rationale for using an overexpression approach (page 4, lines 83-90).

Regarding the limitations of the overexpression approach, we agree with the reviewer that overexpression of the BAG3 fusion protein will increase the probability of aggregate formation, which will accelerate the manifestation of the phenotype. Nevertheless, we don't think that the phenotype is nonspecific as our mouse model quite precisely mimics the human pathology, such as restrictive cardiomyopathy and skeletal muscle atrophy. Of note, restrictive cardiomyopathy in particular is specific to MFM6 and is very rarely observed in mouse models for cardiac diseases, making its coincidental occurrence highly unlikely. Furthermore, higher levels of BAG3^{P209L}-eGFP lead to increased protein accumulation and are therefore comparable to MFM6-patients in later stages of the disease. Although studies of MFM6-patients show no increase in the soluble fraction of BAG3 protein in heart or skeletal muscle, they clearly show the formation of BAG3-positive protein aggregates (Ruparelia et al. 2021 Autophagy and Schänzer et al., 2018 Molecular Genetics and Metabolism) as seen in our mouse model.

The authors do not underline that very high levels of expression hBAG3P209L with an eGFP tag further suggests that it may have a larger effect on aggregating of the targets of Bag3 and wildtype Bag3.

Answer: We thank the reviewer for raising this important point. To address this, we have included new data from a control mouse line overexpressing human BAG3^{WT}-eGFP. This mouse line serves as an ideal

control, as it was generated in the same way as the CAG-BAG3^{P209L}-eGFP mouse line (knockin in the Rs26 locus), with the only difference being that the human BAG3 does not carry the P209L point mutation. We now show that there is a comparable induction of BAG3 mRNA in skeletal muscle of both transgenic mouse lines, CAG-hBAG3^{P209L}-eGFP and CAG-hBAG3^{WT}-eGFP (control), while the total BAG3 protein (mouse + human) is 11.36-times increased in CAG-hBAG3^{P209L}-eGFP-mice but only 4.46-times in CAG-hBAG3^{WT}-eGFP-mice. This is due to the accumulation of hBAG3^{P209L}-eGFP protein in aggregates, which is not seen in the control mouse line.

Regarding the ratio of overexpressed to endogenous BAG3, there is 1.54 times more BAG3^{WT}-eGFP protein in skeletal muscle of the CAG-hBAG3^{WT}-eGFP mouse line than mouse Bag3 protein. This is quite similar to the CAG-hBAG3^{P209L}-eGFP mouse line, which has 1.92 times more BAG3^{WT}-eGFP protein than mouse Bag3 protein. We have added a new paragraph discussing the differences in expression levels of protein and mRNA for mouse Bag3, human BAG3^{WT}-EGFP, and BAG3^{P209L}-eGFP in our transgenic mouse lines (page 6, lines 133-155).

More discussion on the relative tissue expression of the wildtype and hBAG3P209L should have been included to insist on the difference between heart and skeletal muscle.

Answer: We have included expression data comparing BAG3^{P209L} protein levels in heart and skeletal muscle (page 6, lines 145-150) and added a short discussion of the relative tissue expression to the discussion (page 28, lines 624-629). The total amount of BAG3 protein (human BAG3^{P209L} and mouse Bag3) in the hearts of these mice is 2.6 times higher than in skeletal muscle. This explains why the phenotype is more severe in hearts.

A model with at the most a hBAG3P209L expression level comparable to the wildtype would have been preferable.

Answer: We agree with the reviewer and present a control line in which hBAG3-eGFP protein is overexpressed to an extent comparable to the hBAG3^{P209L}-eGFP protein (ratio of hBAG3^{WT}-eGFP to mBag3 is 1.54 compared to a ratio of 1.92 for hBAG3^{P209L}-eGFP to mBag3). Importantly, no obvious phenotype is observed in this control mouse line. Therefore, we conclude that our mouse model, despite potential drawbacks (overexpression of a non-physiological amount of the hBAG3-eGFP fusion protein), is still a valid model mimicking MFM6, as underscored by the observed phenotype.

Questions to be addressed

Page 13: How significant was the "...increase of BAG3, αB-crystallin (Fig. 5c, d) and the small heat shock proteins HSPB7 and 8 (Fig. 5e, f), of which HSPB8 binds BAG3" should be stated in this sentence.?

Answer: We have added the word "statistically" to indicate a statistically significant increase of the mentioned proteins (page 16, line 361).

Page 17: They provide strong evidence that overexpression of hBAG3P209L leads to the: "... 215 of proteins accumulating in the pellet fraction ($\log_2(\text{BAG3P209L}/\text{control}) > 0.25$) showed reduced abundance in the soluble fraction ($\log_2(\text{BAG3P209L}/\text{control}) < -0.25$), indicating increased association to the cytoskeleton or aggregation (Fig. 6d). More data on relative overexpression and phenotype should be included.

Answer: We now include a correlation between the maximal force generation of the EDL and soleus muscles and BAG3^{P209L}-eGFP expression (Suppl. Fig. 4k,l). We show that the higher BAG3^{P209L}-eGFP expression, the lower the maximum force development. However, this does not apply to BAG3^{WT}-EGFP

controls, which indicates that mice with higher BAG3^{P209L}-eGFP expression mimic patients in later stages of the disease with an increased build-up of protein aggregates.

Page 21: Over expression is likely the key driver to phenotype, which is not what is observed in the human pathology, as its level influence severity as stated: “This impairment correlated with the expression level of the BAG3P209L-eGFP transgene and explained the phenotypic variability we observed in this mouse line. The reason for the different expression levels of the transgene are unclear as all mice analyzed were homozygous for the activated BAG3P209L-eGFP expression.”

Answer: We thank the reviewer for raising this point, and would like to explain this in more detail: We consider the level of BAG3 protein expression, as measured by Western blot analysis, to be part of the phenotype. We now provide mRNA expression data for the CAG-BAG3^{P209L} mouse line and demonstrate that the mRNA-level of mouse Bag3 increases only 1.3-fold (a 30% increase, which is not statistically significant), while the amount of mBag3 protein increases 5.6-fold (460% increase). The mRNA level of hBAG3-eGFP is 1.8 times that of mBAG3 in non-transgenic controls. Total mRNA of BAG3 (mouse and human) is ~3-times higher than in non-transgenic CTRLs, but total BAG3 protein is ~11-times higher than in CTRLs. This indicates the accumulation of BAG3^{P209L}-protein, which is part of the phenotype and is only partly due to overexpression. This was further corroborated by analysis of our CAG-BAG3^{WT} mouse line. In this line, we observed an approximately 5-fold increase in total BAG3 protein in skeletal muscle without protein aggregate formation or functional impairment. mRNA-expression of mBAG3 and hBAG3 in these mice was comparable to that of CAG-BAG3^{P209L}-mice (mouse Bag3 mRNA 0.038±0.018 in homozygous BAG3^{P209L}-eGFP mice and 0.054±0.018 in homozygous BAG3^{WT}-eGFP mice and human transgene mRNA was 0.061±0.033 for BAG3^{P209L}-eGFP and 0.034±0.026 for BAG3^{WT}-eGFP (Suppl. Fig. 3a)). From this, we conclude that expression BAG3^{WT}-eGFP protein at levels up to at least 5 times the basal level of endogenous Bag3 can be tolerated without causing protein accumulation or triggering a typical MFM6 phenotype.

Page 22: This difference in expression between heart and skeletal muscle are important. Could we compare levels of expression in the mice?

Answer: As requested by the reviewer, data from Western Blot analysis comparing BAG3/Bag3 expression levels in hearts and skeletal muscle are now included in the revised version of the manuscript new (Suppl. Fig. 3c,d). The total amount of human BAG3 and mouse Bag3 protein is 6.36±0.52 (a.u.) for heart and 2.45±0.96 (a.u.) for skeletal muscle (new Suppl. Fig. 3d). Thus, expression in the hearts of these mice is 2.6 times higher than in skeletal muscle.

Minor points

A supplemental figure with the alignment of the hBAG3P209L compared to the mice Bags3 would be appreciated.

Answer: A new supplemental figure depicting the alignment of the amino acid sequences was added to the supplement (new Suppl. Fig. 14) and mentioned in the revised Discussion (page 27, lines 620-622). The identities at the amino acid level are 484/581, which corresponds to 83%.

Page 11 legend of figure 4: 20 most differentially expressed genes are shown. Not genes but pathways.

Answer: In the Volcano plot of Fig. 4a, the 20 most differentially expressed genes are shown. We see no need to change the legend.

Page 12: adherents rather than “adherens”

Answer: We thank the reviewer for this suggestion. However, after studying various publications, we come to the conclusion that “adherens” rather than “adherents” junctions is the correct term for this kind of cell-cell contact.

Page 16: Figure d, the lines to dots for hBAG3P209L and mice Bag3 are not clear.

Answer: We have changed Fig. 6d accordingly: The lines pointing to the dots have been made thicker. The “Bag3” labeling has been moved so that it can no longer be confused with the labeling of the hBAG^{P209L}-eGFP.

Reviewer #3 (Remarks to the Author):

The authors present a mouse model for BAG3P209L myofibrillar myopathy that shows molecular and functional features of severe muscle weakness as described in patients with MFM6. They demonstrate that accumulation of BAG3-containing aggregates and loss of BAG3 function leads to sarcomere disintegration, mitochondrial dysfunction and subsequently muscle weakness and this can be rescued by knockdown of hBAG3P209L.

While potentially interesting, the study is essentially uncontrolled because they did not generate and characterize a mouse line that overexpresses WT BAG3 to the same level. Overexpression of many WT proteins leads to phenotypes so there is no way here of knowing if the observed phenotypes were due to the P209L mutation or gross overexpression of BAG3.

Answer: We appreciate Reviewer #3’s critical observation regarding the need for a control mouse line that overexpresses a fusion protein of human BAG3^{WT} and eGFP. We have generated a transgenic mouse line that overexpresses human BAG3^{WT}-eGFP under the control of the ubiquitous CAG-promoter. Since the transgenic construct was knocked into the Rs26 locus, is inducible, and was crossed with a PGK-Cre mouse line and bred to homozygosity, it provides the ideal control, as requested, to demonstrate the specificity of BAG3^{P209L}-eGFP overexpression in mimicking MFM6. To strengthen our conclusions, we have conducted key experiments with this hBAG3^{WT}-eGFP mouse line and did not observe any obvious phenotype (new Suppl. Figs. 1, 3, 7). From these experiments we can conclude that the phenotype observed in CAG-BAG3^{P209L} mice is not an artifact caused by uncontrolled overexpression of a human BAG3^{WT}-eGFP fusion protein, but rather a specific manifestation of the MFM6 phenotype.

Reviewer #4 (Remarks to the Author):

In the manuscript the authors studied the role of BAG3 mutation in muscle physiopathology. They generated a humanized transgenic mouse line that ubiquitously expressed the mutant BAG3P209L protein. These mice have been previously characterized from a cardiac point of view and indeed they showed a cardiomyopathy and died within 5-8 weeks. When the authors checked the skeletal muscle function and structure, they found the typical features of Myofibrillar myopathies (MFM) such as myofibrillar disassembly, protein aggregates and weakness. Transcriptomics and proteomics analyses confirmed an alteration in proteostasis with impairment of autophagy/mitophagy and accumulation of dysfunctional mitochondria. Almost all these features were reverted by a genetic approach to knockdown BAG3 expression in skeletal muscles. The paper is primarily descriptive of the muscle phenotype when

BAG3^{P209L} protein is expressed. The phenotype is very interesting but the insights that explain the muscle features with BAG3^{P209L} are superficially explored. The authors should consider the following points

We thank the reviewer for his/her thoughtful criticisms and suggestions. We have performed, as requested, new experiments and these helped to further clarify the disease mechanisms underlying the MFM in our transgenic mouse model.

Point1. Figure 1f-g. The GFP pattern is not homogeneous among the different myofibers, especially in the soleus. Is there any fiber type specificity? From the pattern it looks like that it is higher expressed in the fast fibers. Authors should check the GFP expression in the different fiber types

Answer: We thank the reviewer for raising this point. We have included new data on fiber type in the revised version of the manuscript (Suppl. Fig. 4) and found a switch from fast to slow fibers in soleus and EDL muscles from 4 week old CAG-BAG3^{P209L}-mice. Specifically, the percentage of MHC-I type fibers increased, while the percentage of MHC-IIb fibers decreased. This has been observed in several neuromuscular diseases, such as Duchenne muscular dystrophy (Talbot and Maves, Wiley Interdiscip Rev Dev Biol. 2016, 5: 518–534.).

Interestingly, there was no preference for BAG3^{P209L}-eGFP expression in the different fiber types in the soleus and EDL muscles, as the expression levels were not statistically significantly different. Nevertheless, we observed a very high variability when comparing individual values.

Point2. Authors should perform morphometric analyses in different muscles such as Tibialis Anterior, Gastrocnemius and Diaphragm.

Answer: We thank the reviewer for this helpful suggestion. We do include an analysis of the diaphragm, as the failure of this muscle is one of the main causes of MFM6 fatality (new Fig. 1f,k,l,p; new Fig. 2a,d; new Fig. 3d; new Suppl. Fig. 1b). We found BAG3^{P209L}-eGFP positive protein aggregates, increased fibrosis and a higher number of centralized nuclei.

In addition, our extensive analysis of different skeletal muscle types focused on the soleus, quadriceps, diaphragm, and EDL muscles, because quadriceps and EDL mainly consist of fast twitch type II fibers, and soleus and diaphragm mainly of slow twitch type I fibers, type II fibers, and the quadriceps mostly of a mixed type (quadriceps). With this selection, we believe we have effectively characterized the various skeletal muscle types.

Point3. Fig2a. Check also Diaphragm, which is the most affected muscle in murine models of Muscle Dystrophy. Check also fat infiltration with Sirius red staining

Answer: We examined the diaphragm muscle and now present histological data (see also above). We show homogeneous BAG3^{P209L}-eGFP expression (new Fig. 1f) and the formation of eGFP-positive protein aggregates (Fig. 3d). Histology revealed an increase in centralized nuclei (new Fig. 1k,l,p) in Soleus, Quadriceps, and EDL muscles, indicating muscle pathology. We have also performed Sirius red staining, as requested by the reviewer, on the diaphragm of CAG-BAG3^{P209L}-mice (new Fig. 2a) and report a statistically significant increase in fibrosis (new Fig. 2d). We have not observed fat infiltration with either Sirius red or H & E stainings in the diaphragm, EDL muscles, soleus muscle, or quadriceps muscle.

Point4 Fig3c-d are of poor quality. Show high resolution images

Answer: The images in Fig. 3c-d have been replaced with high-resolution images.

Point5. Fig3e-g. EM pictures suggest accumulation of vesicles/lysosomes. Show high magnification pictures to underline the features. Authors should immunostain for Lamp1, p62 and LC3 and quantify the puncta.

Answer: We have added SQSTM1 and LC3 staining of skeletal muscle from CAG-BAG3^{P209L} and control mice and quantified, as requested by the reviewer, the puncta (new Fig. 3e,f). We observed a statistically significant increase in SQSTM1 and LC3B positive puncta.

Point6. Autophagy and mitophagy flux must be monitored to support the authors claims.

Answer: We have addressed this interesting point with new experiments using primary skeletal myocytes from CAG-BAG3^{P209L} mice. We isolated satellite cells from 4-week-old mice and treated them with Bafilomycin A1 (BAF A1) to inhibit autophagy and mitophagy. As a readout for autophagy, we used SQSTM1, and as a readout for mitophagy, PINK1 stainings. We demonstrate that autophagy and mitophagy are stalled (blocked) in skeletal myocytes from CAG-BAG3^{P209L}-mice, as there was no difference in total amount after BAF A1 treatment (new Fig. 7b,c). Additionally, there is no change in BAG3^{P209L}-eGFP fluorescence intensity, which suggests no degradation of BAG3-protein aggregates (new Fig. 7d). As a positive control for autophagic and mitophagy flux, we overexpressed Beclin1 and BNIP3 in skeletal myocytes and observed a significant increase of SQSTM1 and PINK1, respectively (new Fig.7b,c).

Point7, In vivo Bag3 knockdown results in phenotypic amelioration. Authors should also monitor whether mitochondrial function and autophagy was restored.

Answer: This is addressed in our Western Blot analysis results and functional data: Both mitochondrial function and autophagy are restored, as we see a decrease in mitophagy, as indicated by PINK1 expression (new Suppl. Fig. 7a,m) and a decrease in blockage of autophagy, as measured by LC3B and SQSTM1 expression (Suppl. Fig. 7a,f,l). In addition, muscle force increased significantly (Fig. 8h,i), which is an indicator (although indirect) of improved mitochondrial function.

Point8. Author should put an effort in identifying which mechanisms downstream BAG3 dysfunction is the main trigger of the phenotype. Is autophagy reactivation sufficient to restore muscle function in BAG3^{P209L} mutant? Or Is mitochondria improvement via mitophagy? Or just mitochondrial function that when preserved also improves muscle function. These experiments could be easily done (e.g. Beclin1 expression or rapamycin treatment for autophagy activation, BNIP3 expression for mitophagy, PGC1 α expression for mitochondrial biogenesis and function) and would confirm/identify insights that connect BAG3 mutation to MFM,

Answer: We thank the reviewer for raising the criticism, whether autophagy or mitophagy primarily underlie the pathological phenotype. Therefore, we have performed experiments with primary skeletal myoblasts from the skeletal muscles of CAG-BAG3^{P209L} mice, which we differentiated into skeletal myocytes. As suggested by the reviewer, we transfected primary differentiated myocytes with expression vectors for Beclin1, which activates autophagy, BNIP3, which induces mitophagy, and PGC-1 α , which induces mitochondrial biogenesis. Interestingly, only the induction of autophagy had a significant impact on the phenotype, as there were fewer SQSTM1-positive and BAG3-positive protein aggregates in the cells (new Fig. 7a,b,d). The cells also displayed improved cross-striation and were larger. We measured autophagic flux in these cells by treating them with Bafilomycin A1. Again, an increase in SQSTM1 was observed only in Beclin1- treated cells consistent with the induction of autophagic flux (new Fig. 7b). Mitophagy flux, measured based on PINK1 levels, was increased after BNIP3 overexpression (Fig. 7c), but did not reduce SQSTM1 or BAG3^{P209L}-eGFP levels (new Fig. 7b,d). In addition, sarcomere morphology was assessed using the SarcAsM algorithm (Haertter et al., 2024 eLife)

and revealed a statistically significant increased Z-band length in BECN1 transfected cells compared to CTRL and PGC-1 α transfected cells (new Suppl. Fig. 11a).

Furthermore, we have explored the downstream effects of BAG3^{P209L} by inducing autophagy with rapamycin in CAG-BAG3^{P209L} mice. This led to an amelioration of the phenotype, as shown by increased grip strength, better motor coordination (rota-rod), and more voluntary movement (new Suppl. Fig. 11e-g). Measuring LC3B-II content in the quadriceps muscle of these mice showed an increase in this autophagy marker (new Suppl. Fig. 11h).

We conclude from these experiments that impaired autophagy, not impaired mitophagy is responsible for the skeletal muscle phenotype.

Point-by-point response to Reviewer Comments

Reviewer #2 (Remarks to the Author):

The authors have made very significant modifications to the manuscript, including adding data on a new mice line that further supports that the original line is a good model for myofibrillar myopathy 6 and not due to overexpression of the tagged transgene. They have made very constructive modifications in response to the different reviewers comments that have improved clarity. This is an important contribution to the field.

Answer:

We thank the reviewer for his/her positive comments and his/her appreciation of our work.

Response to Reviewer #3:

Whilst the authors have made a valiant attempt to address my concern that the phenotypes associated with transgenic overexpression of mutant BAG3^{P209L} might simply be due to overexpression and not the mutation, their control transgenic line only expresses WT BAG3 protein to 39% of the level achieved with BAG3^{P209L} so unfortunately I do not think they can rule out simple overexpression as a source of any or all phenotypes.

Answer:

We have a different view from the reviewer on this specific point for the following reasons:

1. mRNA-expression of mBAG3 and hBAG3 in the CAG-BAG3^{WT}-eGFP mice was similar to that in the CAG-BAG3^{P209L}-mice (total BAG3 mRNA induction 3.37 ± 1.77 in homozygous BAG3^{P209L}-eGFP mice and 3.01 ± 1.41 in homozygous BAG3^{WT}-eGFP mice compared to CTRL (Suppl. Fig. 3a)). As previously explained, the high level of hBAG3^{P209L}-eGFP protein results from the pathomechanism, not its cause, as evidenced by the similar levels of mRNA expression and the defective autophagy with subsequent protein accumulation in skeletal muscles, as shown in Figs. 3a-e, 5b, 6g. As expected, we observed the formation of hBAG3^{P209L}-eGFP protein aggregates in the striated muscle of CAG-BAG3^{P209L}-mice over time, whereas this was not seen in our control mouse line, where hBAG3^{WT}-eGFP protein is overexpressed.
2. Additional evidence regarding protein accumulation is shown when comparing the amount of mouse Bag3 protein in the two mouse lines: Although the mRNA level of mBag3 is higher in mice expressing hBAG^{WT}-eGFP (mouse Bag3 mRNA 0.038 ± 0.018 in homozygous CAG-BAG3^{P209L}-mice and 0.054 ± 0.018 in homozygous CAG-BAG3^{WT} mice (Suppl. Fig. 3a)), the protein level is only increased by a factor of 1.76, whereas in CAG-BAG3^{P209L} mice it is increased by a factor of 5.2.
3. Overexpression of the hBAG3^{P209L}-eGFP fusion protein mimics MFM6 in the hearts of our mouse model, specifically restrictive cardiomyopathy (Kimura et al., 2021, Nat Commun., 12:3575.), which is extremely rare in transgenic mouse models (Liu et al., 2016, Front Physiol., 7:629.) as well as in skeletal muscle exhibiting the pathognomonic phenotype.

Thus, all these findings collectively show, as also acknowledged by Reviewer 2, that the phenotype and cell biological results are caused by BAG3 dysfunction and not merely by overexpression of the protein.